
# Cloud Characteristics, Thermodynamic Controls and Radiative Impacts During the Observations and Modeling of the Green Ocean Amazon (GoAmazon2014/5) Experiment

Scott E. Giangrande[1], Zhe Feng[2], Michael P. Jensen[1], Jennifer Comstock[2], Karen L. Johnson[1], Tami Toto[1], Meng Wang[1], Casey Burleyson[2], Fan Mei[2], Luiz A. T. Machado[3], Antonio O. Manzi[4], Shaocheng Xie[5], Shuaiqi Tang[5], Maria Assuncao F. Silva Dias[6], Rodrigo Augusto Ferreira de Souza[7], Courtney Schumacher[8] and Scot T. Martin[9]

[1]Environmental and Climate Sciences Department, Brookhaven National Laboratory, Upton, NY, USA
[2]Pacific Northwest National Laboratory, Richland, WA
[3]National Institute for Space Research, São José dos Campos, Brazil
[4]National Institute of Amazonian Research, Manaus, Amazonas, Brazil
[5]Lawrence Livermore National Laboratory, Livermore, CA, USA
[6]University of São Paulo, São Paulo, Brazil
[7]State University of Amazonas (UEA), Manaus, Brazil
[8]Texas A&M University, College Station, Texas, USA
[9]Harvard University, Cambridge, Massachusetts, USA

*Correspondence to*: Scott E. Giangrande (sgrande@bnl.gov)

**Abstract.** Routine cloud, precipitation and thermodynamic observations collected by the ARM Mobile Facility (AMF) and Aerial Facility (AAF) during the two-year DOE ARM Observations and Modeling of the Green Ocean Amazon (GoAmazon2014/5) campaign are summarized. These observations quantify the diurnal to large-scale thermodynamic regime controls on the clouds and precipitation over the undersampled, climatically important, Amazon basin region. The extended ground deployment of cloud-profiling instrumentation enabled a unique look at multiple cloud regime controls at high temporal and vertical resolution. This longer-term ground deployment coupled with two short-term aircraft intensive observing periods allowed new opportunities to better characterize cloud and thermodynamic observational constraints as well as cloud radiative impacts for modeling efforts within typical Amazon 'wet' and 'dry' seasons.

## 1 Introduction

The simulation of clouds and the representation of cloud processes and associated feedbacks in Global Climate Models (GCMs) remains the largest source of uncertainty in predictions of climate change [Klein and Del Genio 2006; Del Genio 2012]. Collecting routine cloud observations to serve as constraints for the improvement of cloud parameterizations represents an ongoing challenge [e.g., Mather and Voyles 2013], but one necessary to overcome deficiencies in GCM cloud characterizations. Compounding this challenge, cloud-climate feedbacks operate over extended spatiotemporal scales, while cloud behaviors vary significantly according to the regionally varying forcing conditions [e.g., Rossow et al. 2005]. There is



additional demand to observe and model cloud processes and feedbacks across many undersampled regions, including climatically important tropical locations where it is often difficult to deploy ground equipment.

As introduced by Martin et al. [2016a; 2017], the Observations and Modeling of the Green Ocean Amazon (GoAmazon2014/5) Experiment was motivated by demands to gain a better understanding of aerosol, cloud and precipitation interactions on climate and the global circulation. The Amazon forest is the largest tropical rainforest on the planet, featuring prolific and diverse cloud conditions that span 'wet' and 'dry' precipitation regimes. These regimes, and associated variations in cloud types, coverage and intensity from sub-daily to seasonal scales, are interconnected to large-scale shifts in the thermodynamic forcing and coupled local cloud-scale feedbacks [e.g., Fu et al. 1999; Machado et al. 2004; Li and Fu 2004; Fu and Li 2004; Misra 2008]. The inability of GCMs to adequately represent clouds over such a complex and expansive tropical area sets apart GoAmazon2014/5 as an important asset for the improvement of GCM cloud parameterizations and simulations of possible climate change [e.g., Williams et al. 2002; Richter and Xie 2008; Nobre et al. 2009; Yin et al. 2013].

One key component for cloud lifecycle and process studies during GoAmazon2014/5 was the two-year deployment of the Atmospheric Radiation Measurement [ARM; Stokes and Schwartz 1994; Ackerman and Stokes 2003] Mobile Facility [AMF; Miller et al. 2016] 70 km to the west of Manaus in central Amazonia, Brazil [$3^{o}12'46.70''$ S, $60^{o}35'53.0''$ W]. This location was chosen to sample the extremes of the local pristine atmosphere, as well as the effects of the Manaus, Brazil pollution plume. The AMF was equipped to capture a continuous record of column cloud and precipitation characteristics from multi-sensor profiling instrumentation, while routine surface meteorology and flux measurements along with balloon-borne radiosonde measurements provided information on the local thermodynamic state [e.g., Kollias et al. 2009; Xie et al. 2015; Tang et al. 2016]. Deploying such an extended, comprehensive cloud instrumentation suite of this sort is unique to Amazon basin studies and rare within global climate-cloud-interaction studies overall, particularly in the tropics. From this dataset, longer-term composites and statistical perspectives on diurnal to seasonal cloud variability (e.g., cloud development, morphological transitions, precipitation occurrence, and radiative properties) are possible.

The long-term, ground-based measurements during GoAmazon2014/5 were complemented with aircraft-based measurements using the DOE ARM Gulfstream-1 (G-1) aircraft (ARM Areal Facility (AAF), e.g., Schmid et al. [2016]). The G-1 was equipped with instruments for measuring clouds, aerosol, chemistry and atmospheric state [e.g., Martin et al. 2017], which provide additional aerosol and cloud microphysical information that is not readily measured at the surface. These data help with the interpretation of ground-based measurements, while the ground measurements assist when determining the representativeness of these aircraft data.

This GoAmazon2014/5 cloud overview serves as a focused cloud study complement to the campaign overview effort found in Martin et al. [2017] and is outlined as follows. Section 2 introduces the AMF instrumentation and methods used for cloud




classification and composite cloud properties. A two-year summary of the environmental conditions and cloud observations in terms of fractional cloud coverages are presented in Section 3. These observations are segregated according to cloud types and contrasts associated with large-scale Amazon wet and dry precipitation regimes. Section 4 details the observations for individual cloud types and their relative impact on surface energy and fluxes. This analysis includes additional relationships

to campaign aircraft in-cloud observations when available. A brief discussion and summary of the initial cloud insights from the GoAmazon2014/5 deployment are found in Section 5.

## 2. ARM Mobile Facility Cloud Observations

The AMF was deployed in Manacapuru, to the west of Manaus in central Amazonia, Brazil (Fig. 1, herein "T3" site; Martin et al. [2017]). Cloud observations were obtained near-continuously over a period from February 2014 through December

2015. The Amazon region surrounding T3 is often identified as the 'green ocean', in reference to its unique, possibly maritime-like atmospheric conditions [e.g., Williams et al. 2002]. The T3 site is situated nearby the intersection of the large Amazon (Rio Solimões) and Rio Negro rivers (Fig. 1), a region of underlying moisture. As a consequence, T3 and the Manaus region may experience increased cloudiness and unique precipitation cycles as compared to the conditions over the larger Amazon basin [e.g., Oliveira and Fitzjarrald [1993], Silva Dias et al. [2004], Romatschke and Houze [2010], Dos

Santos et al. [2014]]. Collow and Miller [2016] recently showed that the presence of the nearby rivers contributed to spatial variability in the regional radiation budgets around the AMF site. Recent GoAmazon2014/5 work has found a robust relation between column-integrated water vapor and precipitation over the Amazon [Schiro et al. 2016]. Seasonal thermodynamical shifts, as well as additional large-scale sea-breeze front type intrusions into the basin (e.g., Cohen et al. [1995], Alcântara et al. [2011]), promote additional cloud lifecycle complexity and diurnal cycle of precipitation variability (e.g., Burleyson et al.

[2016], Saraiva et al. [2016]). Readers are also directed to complementary GoAmazon2014/5 studies on the large-scale environmental controls on clouds, cloud transitions and precipitation found in Ghate and Kollias [2016], Tang et al. [2016], Collow et al. [2016] and Zhuang et al. [2017]. Our analysis focuses on the T3 site that captured a wide range of shallow to deep cloud conditions, sampled and categorized using multi-sensor AMF methods detailed in this section. Continuous environmental forcing datasets over this region were also supported by domain precipitation estimates available from the

System for the Protection of Amazonia (SIPAM) S-band radar operated at the Ponta Pelada airport ("T1", in Fig. 1).

We characterize cloud and precipitation properties according to seasonal and diurnal cycles that separate the observed cloud characteristics between relatively 'wet' (herein, December through April) and 'dry' (herein, June through September) season behaviors. While transitional months (May, October, and November) are not an emphasis of this study, these months contain

several intense (e.g., updraft strength and rainfall rates) deep convective events in the GoAmazon2014/5 record. Thermodynamic profiling (radiosonde) and environmental forcing datasets as sampled over the T3 location are summarized in Section 2.1. To better anchor cloud properties within these wet and dry regimes, aircraft flight operations during



GoAmazon2014/5 prioritized two Intensive Operating Periods (IOPs: 1 February - 31 March, 2014, and 15 August - 15 October, 2014) as introduced in Section 2.2.

Traditionally, Cloud Fraction (CF) observations are of high interest within the GCM community and for high-resolution climate model evaluation [e.g., Bedacht et al. 2007; Wilkinson et al. 2008]. Cloud breakdowns within our study focus on the diurnal to seasonal controls on these CF estimates. This is accomplished by segregating CF properties according to the results of a cloud-type classification algorithm. The multi-sensor approach and cloud classification methods are described in Sections 2.3 and 2.4. Note, the interpretation of CF estimates and 1D column (or, pencil-beam/soda-straw) CF estimate representativeness is often nontrivial [e.g., Wu et al. 2014]. This study defines CF as the fraction of observations (height-resolved, or over the entire column) within an hour for which the combined profiling sensors identify clouds overhead.

## 2.1. Radiosonde, Surface Meteorology and Large-scale Forcing Dataset Overview

During the campaign, radiosondes were launched over T3 at regular 6-h intervals (1:30, 7:30, 13:30 and 19:30 LT, Vaisala RS-92 radiosondes; ARM [1993]). For the IOPs, one additional radiosonde was launched at 10:30 LT to enhance diurnal coverage. Basic thermodynamic processing was performed following Jensen et al. [2015] to estimate convective forcing parameters such as the Lifting Condensation Level (LCL), Mixed-Layer Height (MLH), Convective Available Potential Energy (CAPE), and Convective Inhibition (CIN). For each of these parameters, surface parcels are defined by the level of the maximum virtual temperature in the lowest kilometer. This represents the most buoyant parcel in the boundary layer and maximizes the calculated CAPE. The MLH is calculated using the definition of Liu and Lang [2010] that determines the MLH from a combination of the gradient of potential temperature and the vertical wind shear using criteria based on the stability of the boundary layer and the presence of a low-level jet. Surface radiative flux estimates for this study follow the radiative flux analysis methods of Long and Ackerman [2000] and Long and Turner [2008]. The clear-sky radiative flux estimates are produced by employing an empirical function fitting approach during observed clear-sky periods. The fitted coefficients from these clear-sky intervals are used to interpolate over cloudy periods, providing a continuous estimate of clear-sky irradiances and quality-controlled cloudy-sky fluxes. Detailed analyses of cloud radiative effects are located in Section 4.

Figure 2 presents the cumulative time-series for the two-year GoAmazon2014/15 dataset in terms of average daily profile values for basic cloud, precipitation, thermodynamical and dynamical observations from multi-sensor ground instruments at the T3 site. These efforts complement previous manuscripts on seasonal variability for cloud conditions over the larger Amazon basin [e.g., Machado et al. 2004]. We observe clear shifts in several quantities associated with the Amazon wet and dry seasons. Our ranges for wet and dry season months, as well as the IOP periods, have been indicated in Fig. 2 for reference on their suitability. The more pronounced shifts during the wet season include increased CF in the mid-to-upper





troposphere (between 3-10 km, Fig. 2a), higher precipitation rates (over these daily integrations) and precipitable water (PW, Fig. 2b), as well as the buildup of relative humidity (RH) profiles through the mid-levels (Fig. 2d). CAPE, CIN and zonal/meridional winds (Figs. 2c, e and f) from radiosondes also illustrate large-scale thermodynamical changes and moisture transport associated with wet, dry and transitional periods [e.g., Li and Fu 2004; Fu and Li 2004]. Although

radiosonde daily maximum values indicate only small seasonal changes in CAPE and CIN, we observe that the transitional periods between the dry and wet seasons typically promote maximum relative CAPE trends coupled with relatively lower CIN and heightened moisture. These are the primary ingredients that promote more frequent and strong convection, provided convection can be triggered [e.g., Machado et al. 2004]. Although areal coverage of deeper convection is generally the largest during the wet season, recent profiler-based studies suggest the strongest storms (in terms of upward vertical air

motion) were often observed towards the end of the dry season and into the transitional period (e.g., Giangrande et al. [2016], Nunes et al. [2016]).

The diurnal variation of atmospheric state is illustrated by Fig. 3 and shows the evolutions for the mean and standard deviation of (a) CAPE, (b) CIN, (c) LCL and (d) MLH separated into dry (red bars) and wet (blue bars) components [e.g.,

Betts et al. 2002]. CAPE increases after sunrise, reaching a maximum near midday, whereas CIN is maximum (largest negative value) overnight and decreases during the day. These behaviors are consistent with development of convection breaking the capping inversion and consuming CAPE. Both CAPE and CIN show a stronger diurnal cycle during the dry season compared to the wet season. The mean LCL increases by approximately 600-800 m from sunrise to the afternoon with larger magnitudes and range during the dry season. The mean MLH also increases by approximately 1 km from sunrise

through the afternoon during the wet season, while during the dry season the increase is about 1.5 km. This increase in MLH is consistent with daytime solar heating. Separating the diurnal cycle into dry (red bars) and wet (blue bars) season components indicates slightly stronger diurnal cycle signatures in CAPE, increased CIN and higher MLH for the dry season (similar to measurements obtained in SW Amazon by Fisch et al. [2004]), with a suppressed diurnal cycle in LCL height.

To better inform the observed cloud system variations over the ARM T3 site from the large-scale environmental condition perspective, Fig. 4 plots the diurnal cycle of the large-scale vertical motion (omega), total advection (sum of horizontal and vertical) of moisture and relative humidity. These large-scale fields were derived from the ECMWF analysis outputs over the entire field campaign constrained using the surface rainfall rate from the SIPAM radar following the variational analysis method of Zhang and Lin [1997], updated with using Numerical Weather Prediction analysis in Xie et al. [2004] and again

as in Tang et al. [2016] for the GoAmazon2014/5 deployment. This variational analysis was performed 3 hourly at 25 hPa vertical resolution over a domain of about 110 km in radius, with the center located at the T1 site (Fig. 1).

The omega field shows strong upward air motion in the middle and upper troposphere during the mid to late afternoon (Fig. 4a). The evening and early morning hours exhibit upward air motion confined below 3-4 km, whereas above that level,





downward air motion is dominant. This downward motion is most pronounced between 0600 and 0900 LT. After sunrise, we observe low-level weak ascending motions and positive advection of moisture (Fig. 4d). Between 4-8 km, we observe dry middle tropospheric conditions in the RH field (Fig. 4g). Similar structures in all fields are found across wet and dry season breakdowns, however, middle and upper level descending motions during the evening and early morning hours are much

stronger during the dry season, suppressing convection during those hours. In addition, the ascending motion between noon and late afternoon is much weaker in the dry season compared to the wet season. The dry season also exhibits reduced low-level positive moisture advection and a much dryer lower and middle atmosphere.

### 2.2. The AAF Aircraft Dataset

The DOE AAF G-1 aircraft participated in two IOPs that coincided with the AMF deployment. Airborne measurements were

conducted during 22 Feb. - 23 Mar. 2014 and 6 Sept. - 4 Oct. 2014 representative of the wet and dry season, respectively. The G-1 flight patterns were designed to sample shallow and growing cumulus convective clouds that formed downwind from Manaus to examine the evolution of urban pollution and its effect on cloud and precipitation properties [Martin et al. 2017]. Typical flights consisted of a series of level legs flown just below cloud base, just above cloud base, and higher in growing cumulus clouds, including legs over the T3 ground site. In total, sixteen and nineteen flights in warm cumulus

clouds were included in the wet and dry season, respectively.

The G-1 payload was designed to measure the full spectrum of aerosol size from 0.015 µm to 3 µm and cloud particle sizes from 2 µm to 1.92 cm. For this study, three cloud particle distribution probes are combined to create the full drop-size distribution (DSD) depictions presented in Section 4. The Droplet Measurements Technologies Cloud Droplet Probe (CDP;

2-50 µm) is combined with the Spec Inc. 2-Dimensional Stereo probe (2-DS; 10 µm - 3 mm) between 20 and 50 µm by averaging the overlapping bins. The Spec Inc. High-Volume Precipitation Spectrometer (HVPS; 150 µm – 1.92 cm) is used for droplets larger than 500 µm. Cloud droplet distributions are combined by averaging the DSD for each instrument separately over these flight periods. This was done for in-cloud conditions only. DSDs from the CDP are used for drops smaller than 20 µm. The DSDs from the CDP and 2-DS are averaged between 20 and 50 µm, 2-DS DSDs are used between

50 and 500 µm, and HVPS DSDs are used for drops larger than 500 µm. The 2-DS probe occasionally contained artifacts known as 'stuck bits', i.e., when a photodiode becomes continuously occulted due to optical contamination or electronic noise [Lawson et al. 2006]. Each flight was visually inspected for artifacts, which were manually removed from the combined DSDs. Cloud condensation nuclei (CCN) were measured with a dual-column system manufactured by DMT (operated with a constant pressure inlet at 600 mbar), and Liquid Water Content (LWC) was measured using a multi-wire

element probe (Science Engineering Associates (SEA) Water Content Meter WCM-2000) with wire sizes the same as King and Johnson-Williams probes.



### 2.3. Radar Dataset and Multisensor Merging

The ARM 95-GHz W-band ARM Cloud Radar (WACR) [e.g., ARM 2005] is the primary profiling instrument to characterize the cloud conditions during GoAmazon2014/5. Cloud masking and designation products are performed using the multi-sensor WACR preprocessing approach following Active Remote Sensing of Clouds methodologies [ARSCL; Clothiaux et al. 2000; Kollias et al. 2005, 2009], and additional quality control refinements following Kollias et al. [2014]. These retrievals merge observations from the WACR and a collocated laser ceilometer, micropulse lidar (MPL), and microwave radiometer (MWR) to better identify cloud boundaries in the vertical at high temporal (~10s) and vertical (~24m) resolution.

There are several limitations when designating cloud boundaries and hourly CF observations from vertically pointing cloud radars beyond the capabilities of single radar platforms or ARSCL methods [e.g., Lamer and Kollias 2015; Oue et al. 2016]. Primary among these is that the WACR experiences attenuation in rain that manifests as erroneously low or missing cloud top boundaries [e.g., Feng et al. 2009, 2014]. To lessen these impacts within this Amazonian deployment that favors frequent precipitating cumulus, a collocated and well-calibrated 1290-MHz UHF radar wind profiler (RWP; 8-degree beamwidth, 200-m gate spacing, 6-s temporal resolution) was co-gridded to improve cloud coverage through deeper precipitating clouds [e.g., ARM 2009; Giangrande et al. 2013, 2016]. For this study, a modification to the ARSCL cloud boundary designation is produced by merging RWP profiles (operating in 'precipitation' modes, as also described in Tridon et al. [2013]) during precipitation intervals following similar ARSCL-type cloud profile processing [Feng et al. 2014]. The substitution is accomplished using collocated surface rain gauge datasets to help define appropriate "precipitation periods". These are defined as continuous time periods when the surface rain rate from the gauge exceeds 1 mm h$^{-1}$. During these intervals, if more than 10% of the derived WACR first echo-top heights associated with these precipitating clouds is found to be 500 m or more below the echo-top height as recorded by the RWP, a WACR attenuation flag is assigned and the RWP profiles and boundaries are inserted.

Figure 5 illustrates an example of the composite cloud designation for the 1 April 2014 event. Earlier during this event, both the WACR (Fig. 5a) and RWP (Fig. 5b) struggle to sample the thin and/or high cloud regions observed by ARSCL (Fig. 5c). CF estimates in these regions benefit from the additional ceilometer and MPL observations (not shown in Fig. 5) to detect clouds. Congestus clouds, including those having cloud tops ~6 km, are observed reasonably well by both radars. In these times, surface precipitation is limited (Fig. 5d). A deep convective cloud system passes over T3 between 1500-1900 UTC. Heavy precipitation (surface measured rain rate > 60 mm h$^{-1}$) is associated with extinction of the WACR signal, whereas the RWP is able to reconstruct cloud boundaries up to 13 km. Additional precipitation periods are also identified by red bars on top of Fig. 5c, highlighting locations where the cloud boundary designation within precipitation is improved over traditional ARSCL methods.



### 2.4. Cloud Classification and Radiative Properties

A simple cloud-type classification is performed on the cloud boundary and masking dataset from Section 2.3. This approach follows McFarlane et al. [2013] and Burleyson et al. [2015]. These methods classify clouds into seven categories according to the height of the cloud boundaries and cloud thickness. The cloud categories include: shallow, congestus, deep convection, altocumulus, altostratus, cirrostratus/anvil, and cirrus (definitions summarized in Table 1). Figure 5c provides an example of the cloud classifications for 1 April 2014. Cloud classification is used to separate surface radiative properties among the different cloud types. To accomplish this, the nearest cloud profile is matched to the 1-min surface radiative flux data. As with Burleyson et al. [2015], the lowest cloud type present in the column during that time is used to designate the shortwave and longwave radiative flux measurements (Figs. 5e and f) for that cloud type.

### 3. Profiling Observations of Clouds and Precipitation During GoAmazon2014/5

As highlighted in Fig. 2, thermodynamic and cloud properties from this two-year Amazon dataset are diverse and sampled near continuously by the ARM instrumentation to provide unique constraints towards model improvement. First, T3 cloud observations will be summarized according to diurnal and seasonal breakdowns that follow from large-scale shifts between wet and dry Amazon precipitation regimes. Breakdowns of CF associated with each cloud category defined in the previous section are located in Table 2. For composite CF summaries presented in this section, we capitalize on the high temporal and vertical resolution of the ARM instruments to partition CF according to hourly profile estimates.

Measureable precipitation (> 1 mm) was frequent over the T3 site during the campaign according to surface rain gauge observations (as highlighted in Fig. 2b). In this dataset, 216 days recorded measureable precipitation from multiple ARM gauge and radar sensors, with eighty additional days recording light / trace precipitation (< 1 mm). The total campaign precipitation over T3 was approximately 3000 mm. This total T3 accumulation is representative of the regional SIPAM estimates reported in Zhuang et al. [2017], accounting for uncertainty in radar-based rainfall estimates, dataset gaps and discrepancies between point and spatial rainfall estimates. However, this total campaign precipitation may be below normal owing to a late onset of the 2014-2015 rainy season and other factors (e.g., Marengo et al. [2017]). Using collocated RWP echo classification methodologies when available (as described by Giangrande et al. [2016]), it was possible to designate the fractional precipitation associated with convective and stratiform cloud regimes. For this dataset, ~76% of the accumulated precipitation was associated with convective precipitation. Additional details on diurnal and regime breakdowns follow in the subsequent sections.



### 3.1 Cloud and Precipitation Diurnal Cycles

Figure 6 shows diurnal CF profile breakdowns for each cloud category. The 'cirrus' and 'shallow' cloud categories are combined into a single panel since these cloud definitions do not overlap in altitude. Seasonal variations in the diurnal CF by cloud category are described in the next section. Figure 6a indicates that cirrus clouds are the most commonly observed
clouds during the afternoon and overnight hours, whereas shallow cloud observations dominate the early morning hours after sunrise. Combining Fig. 6a with summary cloud occurrence values in Table 2, shallow cumulus in the Amazon are observed with relatively high frequency throughout most of the day (~ 22%). Shallow clouds in the early morning align with low-level weak ascending air motions (Fig. 4a) and the positive advection of moisture (Fig. 4d).  The most common cirrus clouds locations correspond to relatively high RH regions in the upper atmosphere seen in Fig. 4g, where the air is close to
saturation with respect to ice (not shown).

Congestus (Fig. 6c) and deep convective (Fig. 6d) clouds are prominent during the mid-to-late afternoon, with peak CF coverage around local noon. Deeper clouds that include organized convective systems (identifiable using SIPAM observations) appear to maintain higher CFs into the overnight hours (associated with trailing stratiform regions). Integrated
column behaviors are similar to those found from satellite over Manaus from Machado et al. [2004]; specifically, cloud coverage is high throughout the day, peaking after local noon and associated with increased cirrus (Fig. 6a), cirrostratus (Fig. 6b) and deeper convective clouds (Fig. 6d). The T3 location exhibits a pronounced diurnal cycle associated with deeper convection (as also in Saraiva et al. [2016]). This pronounced behavior is further representative of the fortuitous placement for the T3 AMF site, wherein daily cloud lifecycles also phase well with rare propagating sea breeze intrusions over this
portion of the Amazon basin [Burleyson et al. 2016]. Diurnal behaviors of these clouds (enhanced afternoon convection and reduced overnight convective development) align with mid- and upper-level upward motion in the afternoon and downward air motion from night to the early morning (Fig. 4a).

Congestus and altocumulus exhibit weak secondary peaks in the pre-dawn hours (around 0500 LT). This is observed
primarily as a wet season congestus behavior, possibly comparable to suggestions in previous Manaus diurnal rainfall efforts [e.g., Machado et al. 2004]. However, this contribution would typically be dwarfed when combining the rainfall contributions from other cloud types. We note that the non-precipitating categories of altocumulus, altostratus and cirrostratus (Figs. 6b, e, and f) share similar diurnal phasing with cirrus clouds. Cirrus and cirrostratus are more commonly observed than alto-cloud designations. However, we have not differentiated the contributions to cirrus CF estimates that
reflect deep convective or anvil cloud components from other cirrus clouds. Overall, inspection of large-scale forcing fields supports that the diurnal cycle of high-level clouds is not well associated with the diurnal cycle of the large-scale dynamic and thermodynamics. This is indicative of clouds that originate from anvil remnants from deeper convective clouds that developed locally or advected from elsewhere.





Figure 7 shows the diurnal cycle of precipitation properties at the T3 site as observed by the surface gauges collocated with RWP observations. These plots include wet, dry and transitional season contributions (although we do not isolate these transitional months). Average precipitation rates reflect the average across precipitating and nonprecipitating days, peaking

around 1200 to 1600 LT (Fig. 7a), consistent with the deep convective CF in Fig. 6d. Note, the mean rainfall rates including only times when precipitation is present (Fig. 7b) are more comparable between wet and dry season events, suggesting T3 results in Fig. 7a primarily reflect the additional frequency of convection during the wet season and not its relative intensity. Rainfall rates have also been separated into convective and stratiform types as designated by the RWP [Giangrande et al. 2016]. For the composite campaign (dashed line in Fig. 7c), convective precipitation is dominant at ~76% of the fractional

accumulation (~2300 mm) with a relatively flat contribution diurnally (esp. during the dry season). Since this fractional accumulation is based on RWP estimates for convective fraction, it may tend to maximize convective precipitation fraction over traditional scanning radar-based retrievals (e.g., Steiner et al. [1995]). This is because unlike basing these designations on radar reflectivity factor properties and buffering, profiler methods also distinguish columns with convective vertical air motions (including those from elevated sloping updrafts that extend into transition or stratiform regions), as well as shallow

congestus cloud precipitation as 'convective'. Stratiform precipitation (approx. 700 mm) is more frequent (in terms of accumulation) during the overnight hours (30-60%) and associated with the trailing precipitation regions from the convective systems.

### 3.2 Seasonal Cloud Regime Cycles

Figure 7 also plots the diurnal breakdowns for average rainfall rate and fractional convective/stratiform accumulations

during the two wet and dry seasons over T3. Our dataset contains 103 wet season days responsible for approximately 1600 mm of precipitation, and 52 dry season days responsible for approximately 600 mm of precipitation. Thus, wet season months are associated with a factor of 2 increase in average rainfall rates, but even larger increases occur during daytime hours with much smaller changes during the late evening and early morning (Fig. 7a). However, relative to those days having precipitation (Fig. 7b), the differences in the mean rainfall rate are less pronounced, implying dry season convection

as stronger (instantaneously), since the overall convective cell coverage is also reduced during the dry season (e.g., Giangrande et al. [2016]). Wet and dry season convective rain fractions have similar values (~80%) throughout most of the day (Fig. 7c) and only diverge during the early morning hours when wet season convective rain fractions drop to as low as 20%. These diurnal patterns suggest that organized systems pass over T3 primarily in the morning hours during the wet season, but are infrequent and only have a small impact on the multi-month mean statistics.

Figures 8 and 9 plot seasonal breakdowns for the CF diurnal cycle from Fig. 6 for the wet and dry seasons, respectively. Pronounced CF profile increases are associated with deep convective and congestus clouds during the wet season, with the





dry season having less organized cloud contributions [Ghate and Kollias 2016]. Overnight and/or pre-dawn deep convection (e.g., organized, continuation) and additional local congestus development are more common in the wet season. The distinct nighttime enhancement in stratiform precipitation (identified as "Deep Convection" in Figs. 6d, 8d and 9d) during the wet season is consistent with previous findings that propagating convective cloud systems contribute to the observed diurnal

cycle of deep convection [e.g., Burleyson et al. 2016, Tang et al. 2016]. Early morning shallow cumulus CF profiles (Figs. 8a and 9a) indicate frequent low clouds during wet and dry seasons, consistent with a response to increased surface heating and an increase in the surface latent heat flux, with the wet season reporting additional shallow cloud development throughout the diurnal window (and the dry season consistent with elevated LCL heights). Two separate vertical peaks of shallow cumulus CF were observed between predawn and early morning hours (0300-0900 LT) during the wet season: one

right above the surface, and the other at 2 km height. The surface peak is possibly associated with overnight fog being lifted with surface heating associated with the rising sun [e.g., Anber et al. 2015], while the elevated peak may be associated with radiative cooling of the residual boundary layer overnight. Cirrus CF stays elevated during the wet and dry seasons, however cirrostratus/anvil CFs are substantially reduced during the dry season (cf. Figs. 8b and 9b). This pattern suggests mostly a local deep convective contribution to cirrostratus/anvil during the dry season, with local convection and potentially some

additional remnant anvil or decaying organized cloud components advected over T3 during overnight hours under wet season conditions. Quantitative interpretation for these behaviors is challenging owing to coupled cirrus-shallow cloud sampling factors during the overnight hours. For example, it is likely cirrus sampling is shielded (results stemming from an MPL detection) during the wet season owing to the added presence of lower-level clouds and higher relative humidity / attenuation limiting the usefulness of the cloud radar. In contrast, clear low-level conditions during the overnight hours of the

dry season would likely promote improved designation of cirrus. In this regard, wet and dry season cirrus cloud contrasts may be more pronounced than reported by this study.

As highlighted by Figs. 2 and 3, wet season thermodynamical conditions typically favor weaker CAPE, weaker CIN and higher RH in the lower to mid-atmospheric levels, while dry seasons feature stronger CAPE, stronger CIN and lower RH at

the same levels. As inferred from the large-scale forcing fields in Fig. 4, wet season conditions favor higher column relative humidity, as well as heightened moisture convergence throughout the profile. The wet season also features more favorable omega fields at mid-levels for shallow to deeper convective cloud transitions. This behavior is not surprising and also consistent with forcing datasets being constrained using mean domain precipitation estimates. However, as with first year GoAmazon2014/5 studies (e.g., Collow et al. [2016]), only weak correlations are found between cloud behaviors and

thermodynamic parameters (not shown). Nevertheless, coupled thermodynamical and environmental forcing conditions from Section 2 support these observations of more frequent cloudiness during the wet season, visible across all cloud categories when comparing Fig. 8 to Fig. 9.





## 4. Cloud Type Influence on Surface Energy and Fluxes

AMF instrumentation provides unique capabilities to characterize the variability of clouds and their impact on the Amazon surface energy budget (e.g., Collow and Miller [2016]). Previously, Burleyson et al. [2015] quantified the diurnal cycle of surface cloud radiative effects (CRE) over the three ARM sites in the Tropical Western Pacific (TWP, e.g., Long et al.

[2016], ARM [2013]) using long term measurements of ARSCL cloud profiles and surface radiative flux analysis. CRE is defined as cloudy-sky downwelling flux minus clear-sky downwelling flux. By breaking down the aggregate surface CRE by cloud type across the diurnal cycle, Burleyson et al. [2015] found that the largest source of shortwave surface CRE at these three TWP sites comes from low clouds owing to their high frequency of occurrence. Although deep convective clouds have a strong influence on surface shortwave radiation when present, their aggregate impact is limited by a lower frequency of

occurrence compared to shallow cumulus. Compared to SW CRE, LW CRE is typically a factor of 5-6 smaller than SW CRE [e.g., Culf et al. 1998; Malhi et al. 2002, Burleyson et al. 2015]. This study will limit most interpretation to SW CRE. The 2-year deployment during GoAmazon2014/5 allows us to examine the impact of various cloud types on the surface energy budget over the Amazon, providing new details for targeted model improvements of cloud radiative effects in this climatically important, but undersampled region. This deployment also provides a unique opportunity to contrast 'green

ocean' cloud radiative effects during GoAmazon2014/5 with tropical ARM fixed-site measurements in the TWP.

The frequency of occurrence for the lowest cloud types and their associated radiative fluxes (Tables 2 and 3) are composited into hourly bins across the diurnal cycle (Fig. 10). The methodology to produce the radiative fluxes in these tables is similar to Burleyson et al. [2015] to facilitate comparison with previous results over the three TWP sites. We utilize 'as lowest cloud

type' in the column designations in our analysis (e.g., column 2 in Table 2) because clouds closest to the surface typically have the larger impact on the surface radiative fluxes [Burleyson et al. 2015]. However, we also note that it is not possible to separate the radiative impact of multi-layer clouds and sample sizes are potentially too small to only consider single-layer cloud periods. For higher-altitude cloud types, the frequency as lowest cloud in the column is lower than the total cloud frequencies discussed in Section 3 (as reported in column 1 of Table 2). The difference in frequencies is indicative of how

often multi-layer clouds are present (e.g., cirrus clouds are often present above shallow cumulus).

One notable discrepancy with the previous study is that the instrumentation for classifying the clouds that produce significant precipitation (rain-rate > 1 mm h$^{-1}$) during GoAmazon2014/5 is better than the approach used by Burleyson et al. [2015] owing to the merging of the RWP dataset. Specifically, cloud profiles with rain-rate larger than 1 mm h$^{-1}$ are

discarded in Burleyson et al. [2015], but retained for our study. Therefore, we anticipate that cloud radiative effects from precipitating convective clouds (including both congestus and deep convection) are better represented by this study.



### 4.1. Bulk Cloud Radiative Effects

The average aggregated shortwave (SW), longwave (LW) fluxes and CRE measured at the T3 site are given in Table 3, along with long term results from the three TWP sites (Darwin, Manus, Nauru) as reported in Burleyson et al. [2015]. SW CRE dominates (magnitude) as compared to LW CRE. The mean SW CRE (-94.4 W m$^{-2}$) and LW CRE (14.5 W m$^{-2}$)

averaged across the diurnal cycle (nighttime included) over the entire GoAmazon2014/5 is most similar to those found at Manus, which is the cloudiest of the three TWP sites and most influenced by convection in the Western Pacific warm pool. The Darwin, Australia site has a strong monsoonal cycle (i.e., wet/dry season) and the Nauru site is strongly impacted by the El Niño–Southern Oscillation (ENSO) variability [Burleyson et al. 2015]. Manus would be the one most qualitatively consistent with the GoAmazon2014/5 'green ocean' moniker. During the wet season, SW CRE (-115.6 W m$^{-2}$) is twice the

dry season value (-54.9 W m$^{-2}$), although CREs for this region of the Amazon basin are substantially larger than Darwin during all seasons.

Table 2 gives bulk cloud frequency and their radiative characteristics separated by cloud types and by season. The results reveal the averaged reduction of downwelling SW flux when a particular cloud type is present. Consistent with the SW

transmissivity results (Table 2) and those found by Burleyson et al. [2015], congestus and deep convective clouds dominate the conditional (e.g., not a mean property) SW CRE, while cirrus clouds have the smallest effect on downwelling SW flux. Note, the conditional CRE presented in Table 2 includes both single-layer clouds, as well as when additional cloud layers are above the lowest detected cloud layer. This is done deliberately to be consistent with the method used by Burleyson et al. [2015] such that the GoAmazon2014/5 results can be directly compared with their long-term results from the ARM TWP

sites. Examination of the averaged conditional SW CRE calculated using only single-layer clouds reveal a relative reduction of ~26% for altocumulus and ~20% for shallow cumulus clouds, and negligible difference in other cloud types. The reduction in SW CRE when single-layer clouds are considered is likely caused by frequent multi-layer cloud occurrence of cirrus/cirrostratus clouds over shallow cumulus or altocumulus (i.e., artificially inflating the surface SW CRE of cumulus clouds due to additional SW flux reflection by the upper level clouds). The difference in conditional SW CRE between

single- and multi-layer clouds do not change their contribution to the average CRE as discussed below.

### 4.2. Diurnal Cycle of Cloud Radiative Effects by Cloud Type

Comparisons between wet and dry season diurnal behaviors for the frequency of the lowest clouds in the column and the associated mean SW CRE are shown in Fig. 11. Shallow cumulus dominates the SW CRE in both seasons, although their frequency peak two hours earlier during wet season (1000-1100 LT) than during dry season (1200-1300 LT). While the dry

season features reduced frequency and SW CRE of all cloud types, the contrast is most visible for the three convective cloud



types. Shallow, congestus and deep convective cloud mean SW CRE in the wet season are 51%, 69% and 71% larger than that in the dry season, respectively.

### 4.3. Shallow Cumulus Cloud Properties

From the previous section, shallow cumulus (those most frequently observed during the campaign) are associated with large

discrepancies in cloud radiative effects between the wet and dry season (Table 2 and Fig. 11). Further investigation into these clouds and their radiative differences is enabled using aircraft observations available during the GoAmazon2014/5 campaign IOP periods. As discussed in Section 2.2, three cloud particle size distribution probes are combined to create the full DSD (Fig. 12). Combining the cloud microphysical properties in shallow cumulus measured by aircraft observations and the cloud macrophysical properties measured by ground-based instrumentations allow us to explain the cloud radiative effect

differences from wet and dry seasons reported in the previous section.

Cloud particle size distributions (Fig. 12c) in the wet season are characterized by a lesser occurrence of small droplets and a more frequent occurrence of large droplets when compared with cumulus clouds in the dry season. Total number concentration of cloud drops is more than a factor of 2 larger in the dry season versus the wet season (Fig. 12b). However,

the corresponding LWC is roughly the same between the seasons (Fig. 12d). In-situ cloud condensation nuclei (CCN) concentration is also larger in the dry season versus the wet season (Fig. 12a). Aircraft cloud and CCN measurements are consistent with studies that show clouds influenced by aerosol tend to have larger concentrations of smaller droplets and fewer precipitation sized drops for clouds with similar LWC [e.g., Twomey 1974; Cecchini et al. 2016]. Ground-based radar measurements of single layer shallow cumulus clouds at the T3 site show thicker clouds occurring more frequently in the

wet season (Fig. 12e). Likewise, more frequent occurrence of large liquid water path (LWP) from the T3 ground-based MWR in wet season is consistent with the presence of more robust (i.e., vertically developed) shallow cumulus clouds (Fig. 12f). Therefore, shallow cumulus in the wet season is characterized by fewer number but more frequent larger cloud droplets, while those in the dry season is characterized by more frequent smaller cloud droplets (Fig. 12b,c). Interestingly, these differences in DSDs result in comparable LWC between the wet/dry seasons (Fig. 12d). As a result, the stronger

shallow cumulus conditional SW CRE in the wet season, which reflects the difference in microphysical properties, mainly arises from higher values of vertically integrated properties such as LWP and cloud thickness (Fig. 12e,f).




### 5. Discussion, Summary and Future Opportunities

This study documents the continuous observations collected by the DOE AMF and AAF facilities to characterize cloud properties, collocated large-scale environments and cloud radiative effects over the two-year GoAmazon2014/5 campaign. This extended ground deployment included high temporal and vertical resolution cloud profiling instrumentation, enabling a

unique perspective on various cloud types and their diurnal evolution to complement previous satellite-based perspectives over this undersampled region. Routine thermodynamic profiling over the diurnal cycle, targeted IOP aircraft sampling, and collocated aerosol instrumentation support future opportunities to differentiate and interpret cloud lifecycle and process factors influenced by environmental forcing controls and those influenced by coupled cloud-aerosol interactions within pristine and polluted conditions [e.g., Martin et al. 2017].

The propensity for cumulus to initiate, deepen and organize across the Amazon basin drives much of the observed wet and dry season CF profile diurnal contrasts. Amazon wet season environments promote enhanced shallow cumulus throughout the diurnal cycle, as well as additional deeper precipitating cloud development likely associated with reduced CIN, heightened moisture convergence and relative humidity through atmospheric mid-levels. Wet season and transitional periods

exhibiting sharper CAPE and CIN contrasts enhance the likelihood for deep convection to have organized components, thus promoting anvil and trailing stratiform regions that carry into the overnight hours and propagate across the Amazon basin. Weaker secondary peaks in congestus CFs are also found during the wet season within pre-dawn hours, revealed with confidence from coupled ARM profiling observations. Nevertheless, relatively favorable thermodynamical conditions during both seasons supports local congestus to deeper cloud triggering for this ARM dataset, which includes over 200 days

recording measurable rainfall. This regularly occurring daily precipitation is primarily attributed to isolated and locally-driven convective cells, supported by the 76% rainfall accumulation associated with convective modes, as well as the pronounced diurnal cycle for this rainfall centered near local noon. These ideas and T3 representativeness may be further explored using spatial properties as available from collocated SIPAM radar observations during GoAmazon2014/5.

Congestus and deeper convection is also shown to dominate the conditional surface SW CRE, similar to results from previous tropical ARM efforts over the TWP region. As one possible example for the appropriateness of the Amazon 'green ocean' moniker, CRE properties for the Amazon are found to be similar to the TWP ARM Manus location in the Western Pacific warm pool that favors frequent tropical convection with complex influences from adjacent large islands within the Maritime Continent [e.g., Mather 2005]. A natural contrast between Amazon SW CRE behaviors and those from ARM

Nauru observations stems from the strong ENSO-driven variability over this site as a key driver for cloud coverage [e.g., Jensen et al. 1998, Burleyson et al. 2015]. The cumulative Amazon CRE is also larger when compared to the Darwin wet season (given the 'dry' season for Darwin is void of substantial cloud/precipitation). This behavior is partially attributed to



the Darwin monsoonal environments that fluctuate between wider-spread tropical 'Active' cloud conditions and continental 'Break' monsoonal regimes that promote stronger convection [e.g., Holland 1986; May and Ballinger 2007; Giangrande et al. 2014]. Overall, cumulative results from CRE help emphasize the important role of shallow cumulus for the Amazon, including the dry season, and the favorable low-level conditions (e.g., weak ascending air motions, positive moisture

advection and moist surface) throughout the year that promote elevated shallow cumulus frequency. Given this relative importance, these clouds must be properly simulated in both global and regional climate models if the surface radiative budget (that affects land-atmosphere interactions and subsequent convective cloud and precipitation formations over the T3 site) is to be properly represented.

Ground-based multi-sensor measurements and aircraft observations further support thicker clouds occurring more frequently in the wet season. These clouds are those having larger LWP that would also promote the heightened SW CRE and LW CRE contributions. Aircraft and ground-based cloud and CCN measurements and properties for shallow cumulus in this study also informs on the role of the Manaus pollution plume in cumulus cloud evolution. A key motivation behind GoAmazon2014/5 was the opportunity to test various cloud-aerosol interactions in this Amazon natural cloud laboratory setting. Shallow

cumulus summaries provided in our study support prior efforts that suggest clouds influenced by aerosol tend to have larger concentrations of smaller droplets and fewer precipitation sized drops for clouds with similar LWC. As the clean (wet) and polluted (dry) cloud conditions tend to align with large-scale regime thermodynamical controls, subsequent studies will need to differentiate the role of the Manaus plume that influences the observed differences in shallow cumulus microphysical properties, and examine the extent that the reduced frequency for organized precipitation events removes Manaus pollution.

In that regard, impacts on shallow cumulus clouds could have potentially a more profound impact as far as how shallow clouds transition to deeper convection, hence affecting hydrological cycle and land-atmosphere feedbacks.





## 6. Acknowledgements

This manuscript has been authored by employees of Brookhaven Science Associates, LLC under Contract No. DE-SC0012704 with the U.S. Department of Energy (DOE). The publisher by accepting the manuscript for publication acknowledges that the United States Government retains a non-exclusive, paid-up, irrevocable, world-wide license to publish

or reproduce the published form of this manuscript, or allow others to do so, for United States Government purposes. Dr. Zhe Feng at the Pacific Northwest National Laboratory (PNNL) is supported by the U.S. DOE, as part of the Atmospheric System Research (ASR) Program. The PNNL is operated for DOE by Battelle Memorial Institute under contract DE-AC05-76RL01830. Work at the Lawrence Livermore National Laboratory (LLNL) was supported by the DOE ARM program and performed under the auspices of the U.S. DOE by LLNL under contract No. DE-AC52-07NA27344. Funding was also

obtained from the U.S. DOE, the São Paulo Research Foundation (FAPESP - 2009/15235-8), The Amazonas State University (UEA) and the Amazonas Research Foundation (FAPEAM - 062.00568/2014). The work was conducted under scientific licenses 001030/2012-4, 001262/2012-2, and 00254/2013-9 of the Brazilian National Council for Scientific and Technological Development (CNPq). Institutional support was provided by the Central Office of the Large Scale Biosphere Atmosphere Experiment in Amazonia (LBA), the National Institute of Amazonian Research (INPA), the National Institute

for Space Research (INPE), and the Brazil Space Agency (AEB). We also acknowledge the Atmospheric Radiation Measurement (ARM) Climate Research Facility, a user facility of the U.S. DOE, Office of Science, sponsored by the Office of Biological and Environmental Research, and support from the ASR program of that office. All ARM datasets used for this study may be downloaded at www.arm.gov. Additional thanks to Mark Miller (Rutgers University) for an internal review of this manuscript.

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



| Cloud type | Cloud-base height | Cloud-top height | Cloud thickness |
|---|---|---|---|
| Shallow | < 3 km | < 3 km | No restriction |
| Congestus | < 3 km | 3-8 km | ≥ 1.5 km |
| Deep convection | < 3 km | > 8 km | ≥ 5 km |
| Altocumulus | 3-8 km | 3-8 km | < 1.5 km |
| Altostratus | 3-8 km | 3-8 km | ≥ 1.5 km |
| Cirrostratus/anvil | 3-8 km | > 8 km | ≥ 1.5 km |
| Cirrus | > 8 km | > 8 km | No restriction |

**Table 1.** Cloud-type definitions based on cloud boundaries and thickness. Definitions are slightly modified from Burleyson et al. [2015; their Table 2] and McFarlane et al. [2013; their Table 3].





| | | Mean frequency of cloud (%) | Mean frequency as the lowest cloud in column (%) | SW Trans | SW CRE (W m$^{-2}$) | LW CRE (W m$^{-2}$) |
|---|---|---|---|---|---|---|
| Shallow Cumulus | All | 22.3 | 22.3 | 0.51 | -177.0 | 20.5 |
| | Wet seasons | 28.4 | 28.4 | 0.48 | -202.3 | 21.7 |
| | Dry seasons | 16.9 | 16.9 | 0.55 | -128.1 | 17.2 |
| Congestus | All | 5.8 | 4.9 | 0.30 | -329.3 | 29.0 |
| | Wet seasons | 9.0 | 7.6 | 0.29 | -336.8 | 28.6 |
| | Dry seasons | 2.8 | 2.4 | 0.34 | -301.5 | 28.2 |
| Deep Conv. | All | 5.3 | 5.0 | 0.17 | -431.0 | 31.2 |
| | Wet seasons | 9.1 | 8.5 | 0.17 | -418.2 | 29.3 |
| | Dry seasons | 1.5 | 1.4 | 0.17 | -482.1 | 38.7 |
| Altocumulus | All | 20.0 | 13.9 | 0.57 | -130.5 | 13.6 |
| | Wet seasons | 25.7 | 16.2 | 0.54 | -143.6 | 14.5 |
| | Dry seasons | 15.0 | 11.6 | 0.59 | -100.2 | 11.4 |
| Altostratus | All | 2.0 | 1.0 | 0.43 | -232.1 | 19.0 |
| | Wet seasons | 3.2 | 1.5 | 0.46 | -253.5 | 18.1 |





|  |  |  |  |  |  |  |
|---|---|---|---|---|---|---|
|  | Dry seasons | 0.8 | 0.4 | 0.47 | -143.2 | 17.8 |
| Cirrostratus/anvil | All | 7.7 | 4.2 | 0.44 | -242.9 | 14.5 |
|  | Wet seasons | 10.4 | 5.0 | 0.41 | -262.3 | 14.7 |
|  | Dry seasons | 4.2 | 2.6 | 0.50 | -189.1 | 12.4 |
| Cirrus | All | 29.6 | 16.9 | 0.64 | -99.5 | 7.2 |
|  | Wet seasons | 30.8 | 13.6 | 0.59 | -116.9 | 9.7 |
|  | Dry seasons | 25.0 | 18.1 | 0.66 | -81.3 | 5.6 |

**Table 2:** Frequencies of cloud occurrence in the column and associated conditional SW transmissivity (SW Trans), conditional SW cloud radiative effect (SW CRE), and conditional LW cloud radiative effect (LW CRE) for each cloud type. All values are averaged across the diurnal cycle. For SW CRE, only daytime hours are included.



| | SWdn | CSWdn | SW CRE | LWdn | CLWdn | LW CRE |
|---|---|---|---|---|---|---|
| Manaus (Central Amazonia) | | | | | | |
| All data | 197.5 | 291.9 | -94.4 | 420.3 | 405.9 | 14.5 |
| Wet seasons | 181.6 | 297.1 | -115.6 | 423.5 | 405.6 | 17.8 |
| Dry seasons | 227.6 | 282.5 | -54.9 | 414.4 | 405.3 | 9.1 |
| Darwin | | | | | | |
| All data | 232.4 | 293.4 | -61.0 | 407.0 | 394.6 | 12.4 |
| Wet seasons | 226.5 | 321.5 | -95.0 | 427.9 | 411.5 | 16.4 |
| Dry seasons | 239.1 | 262.7 | -23.5 | 384.2 | 376.4 | 7.8 |
| Manus | | | | | | |
| All data | 205.1 | 299.8 | -94.7 | 423.5 | 408.2 | 15.3 |
| Nauru | | | | | | |
| All data | 237.7 | 302.1 | -64.4 | 420.6 | 408.5 | 12.2 |

**Table 3:** Mean downwelling SW radiative flux (SWdn), estimated clear-sky SW radiative flux (CSWdn), aggregate SW cloud radiative effect (SW CRE; SWdn - CSWdn), downwelling LW radiative flux (LWdn), estimated clear-sky LW radiative flux (CLWdn), aggregate LW cloud radiative effect (LW CRE; LWdn - CLWdn). All units are in W m$^{-2}$ and are
5   averaged across the diurnal cycle. The Darwin, Manus, Nauru results are taken from Burleyson et al. (2015), Table 3.





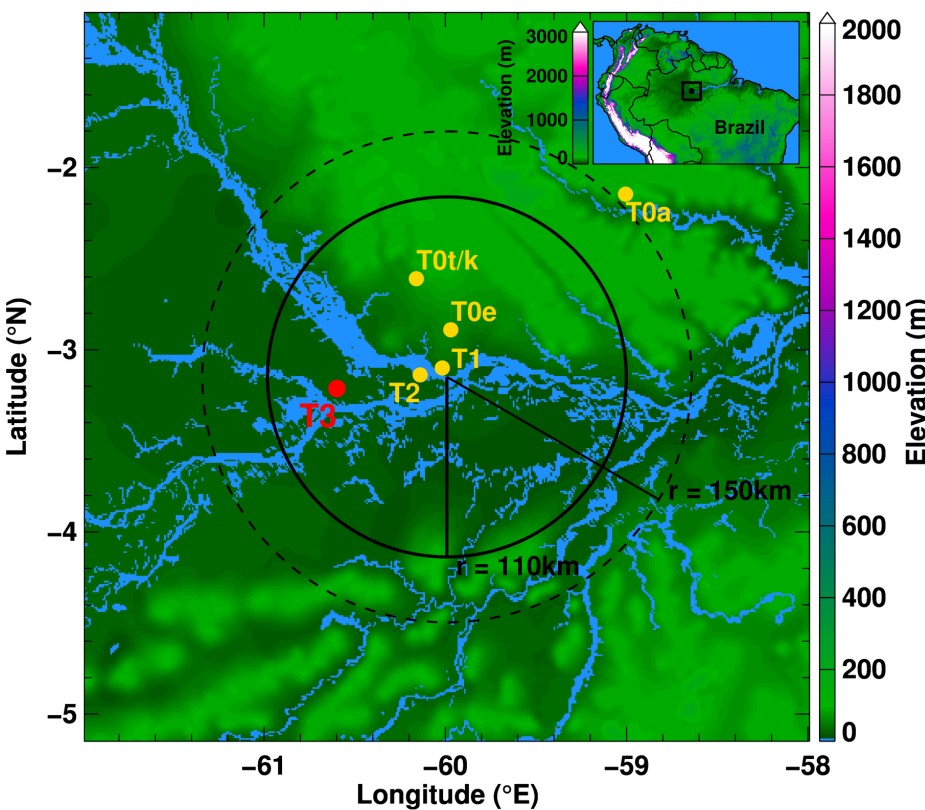

**Figure 1:** Location of the GoAmazon2014/5 key deployment sites and associated terrain elevation (shaded). The primary ARM AMF facilities were located at the T3 location. Range rings indicate distances from the SIPAM radar location near T1. The 110 km range ring is the range associated with the ARM continuous forcing dataset domain.

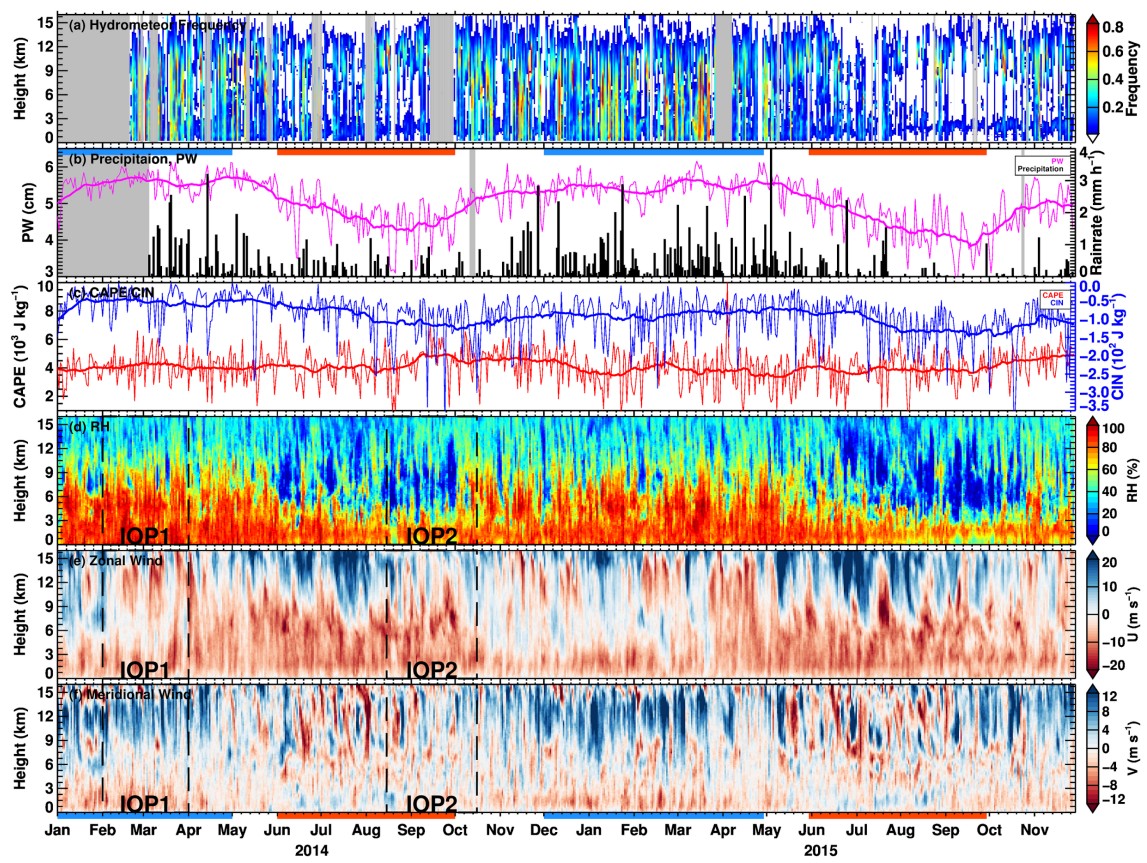

**Figure 2:** Time series of (a) hydrometeor frequency from the merged WACR-ARSCL-RWP dataset, (b) column precipitable water (purple, thick line is 30-day running mean) and surface precipitation (black bars), sounding measurements of (c) daily maximum CAPE (red), daily minimum CIN (blue), (d) relative humidity (with respect to liquid), (e) zonal wind, and (f) meridional wind. The data shown are daily average values. Gray fillings in (a,b) are periods with missing cloud and

10    precipitation data, respectively. 'Wet' and 'Dry' seasons in this study are denoted with blue and orange bars in (b,f), IOP1 and IOP2 periods are denoted by the dash lines in (d-f).





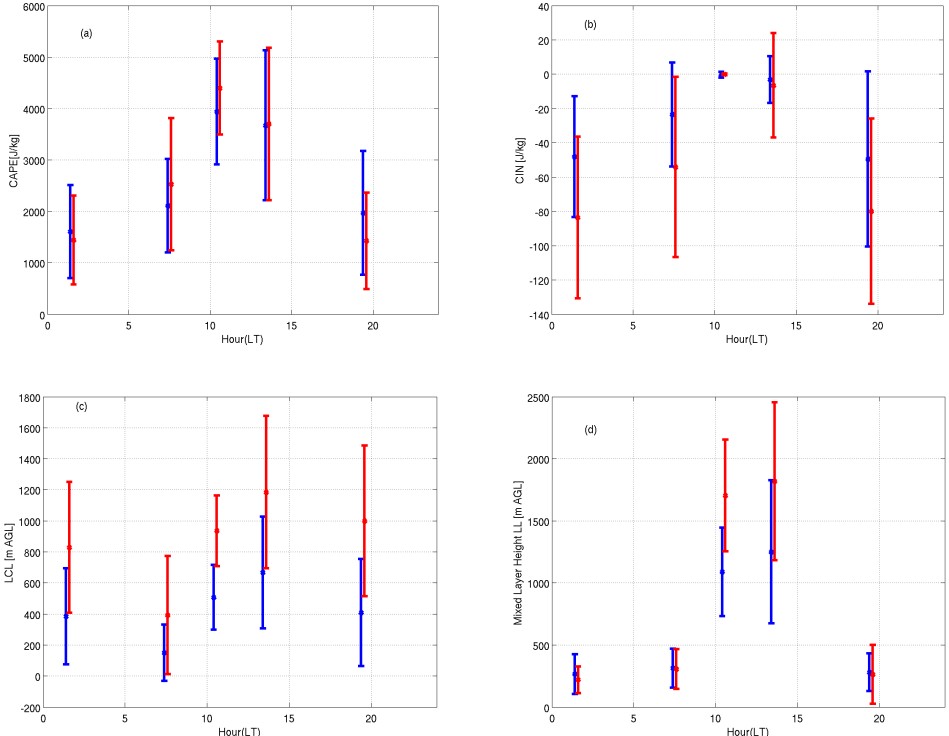

**Figure 3:** Diurnal cycles (mean and standard deviation) for radiosonde-based thermodynamic quantities of (a) CAPE, (b) CIN, (c) LCL height, and (d) MLH for wet (blue) and dry (red) season breakdowns.



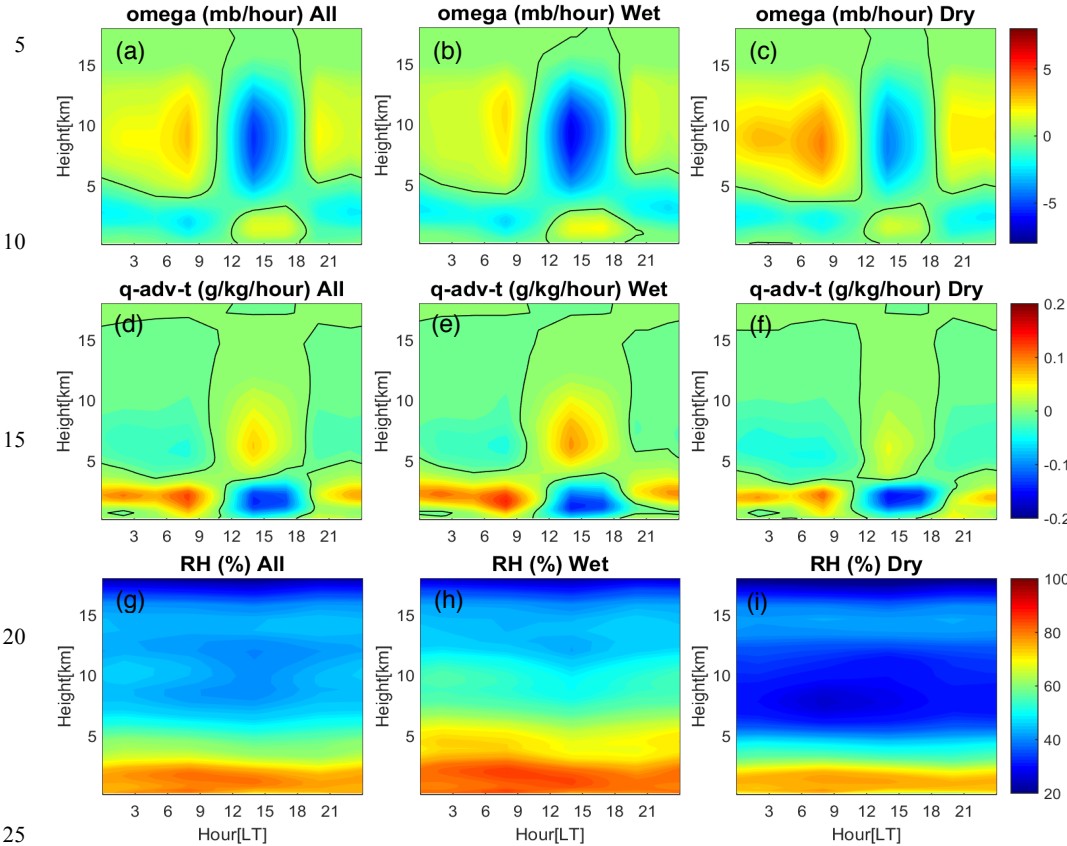

**Figure 4**: Diurnal cycles of omega, total advection of moisture and relative humidity (with respect to liquid) for the complete two-year GoAmazon2014/15 campaign record (left), as well as wet season (middle) and dry season (right) breakdowns.





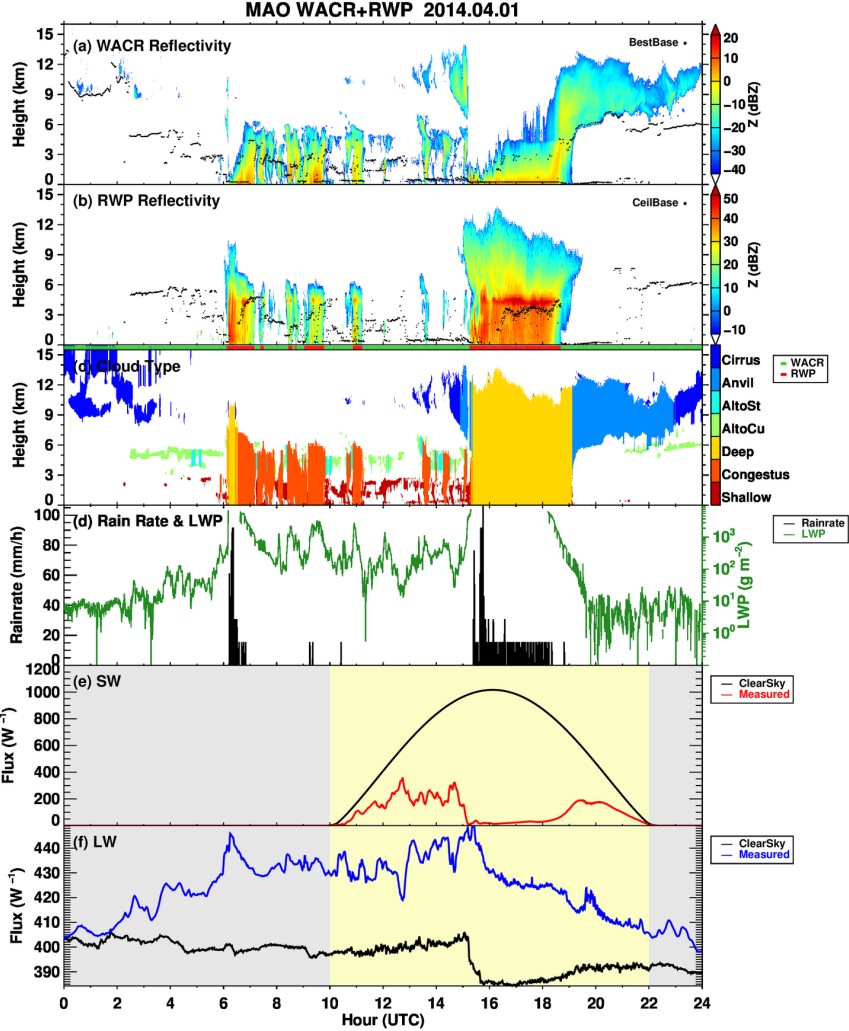

**Figure 5:** Example from the merged WACR-ARSCL-RWP dataset for a 1 April 2014 event: (a) WACR reflectivity, (b) RWP reflectivity, (c) cloud type classification, (d) tipping bucket rain rate (black) and MWR retrieved liquid water path (green), (e) downward shortwave flux, and (f) downward longwave flux. The black dots in panels (a,b) reflect a best estimate cloud-base height and ceilometer cloud-base height from the WACR-ARSCL dataset. The black lines in panels (e,f) reflect clear-sky flux estimates from the radiative flux analysis product.




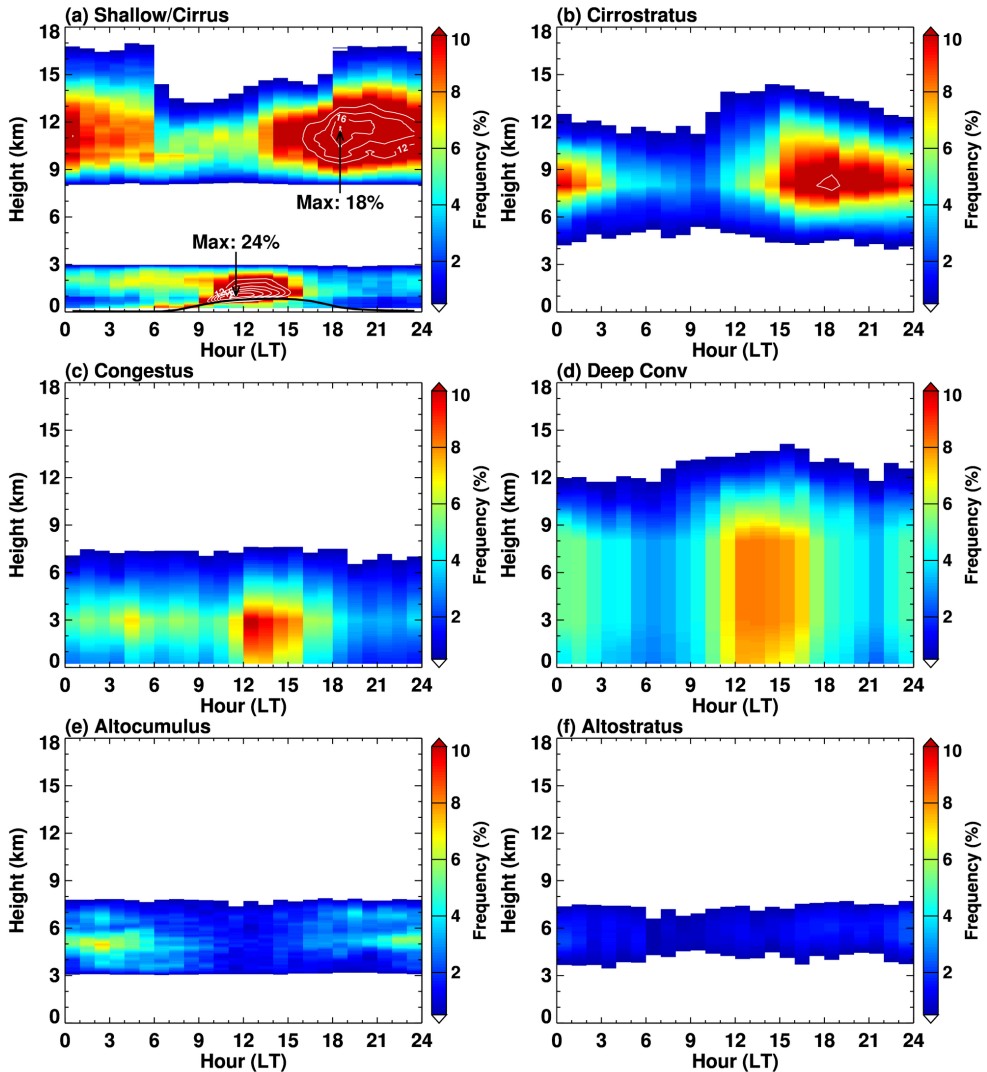

**Figure 6:** Composite diurnal cycle cloud fraction profiles segregated according to each of the seven cloud classification categories. White contours start at 10% and increment at 2%. Maximum cloud fraction values for shallow and cirrus clouds are marked in (a). The black line in (a) plots the averaged LCL height estimated using surface measurements.





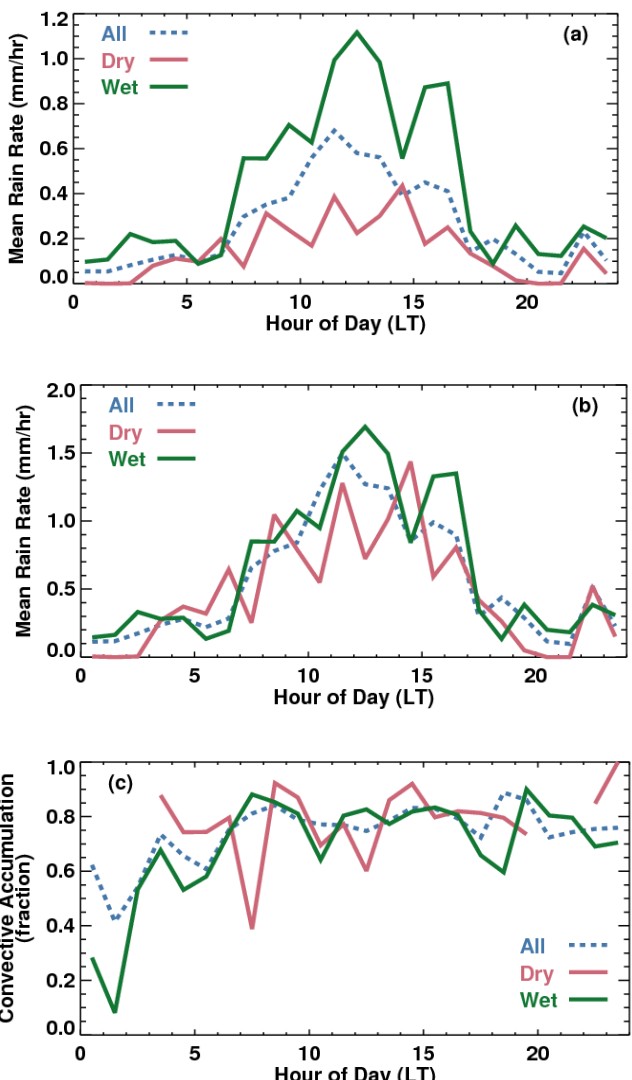

**Figure 7:** (a) Mean daily precipitation rate [mm hr-1] for all days, (b) for only the precipitating days during the campaign (>

5     1 mm hr-1) and (c) the fractional convective accumulation as sampled by the raingauges for summary campaign and

associated wet and dry season conditions.



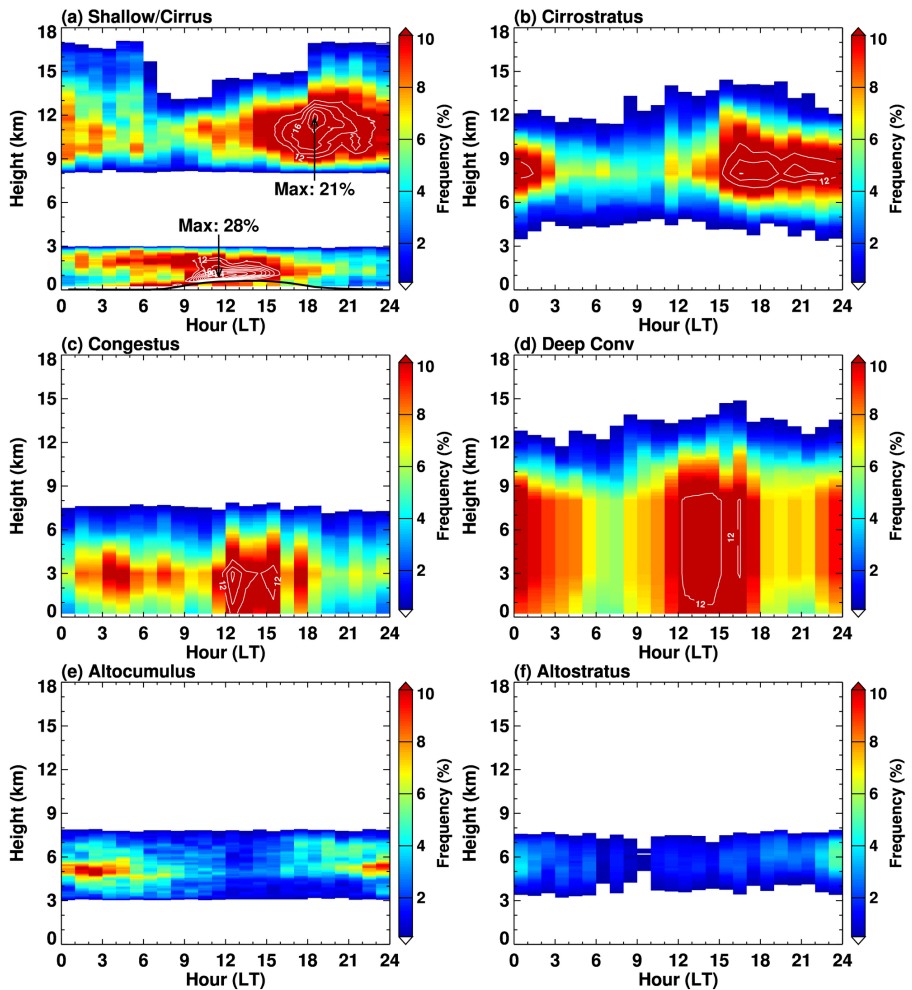

5  **Figure 8:** As in Fig. 6, but for wet season conditions.





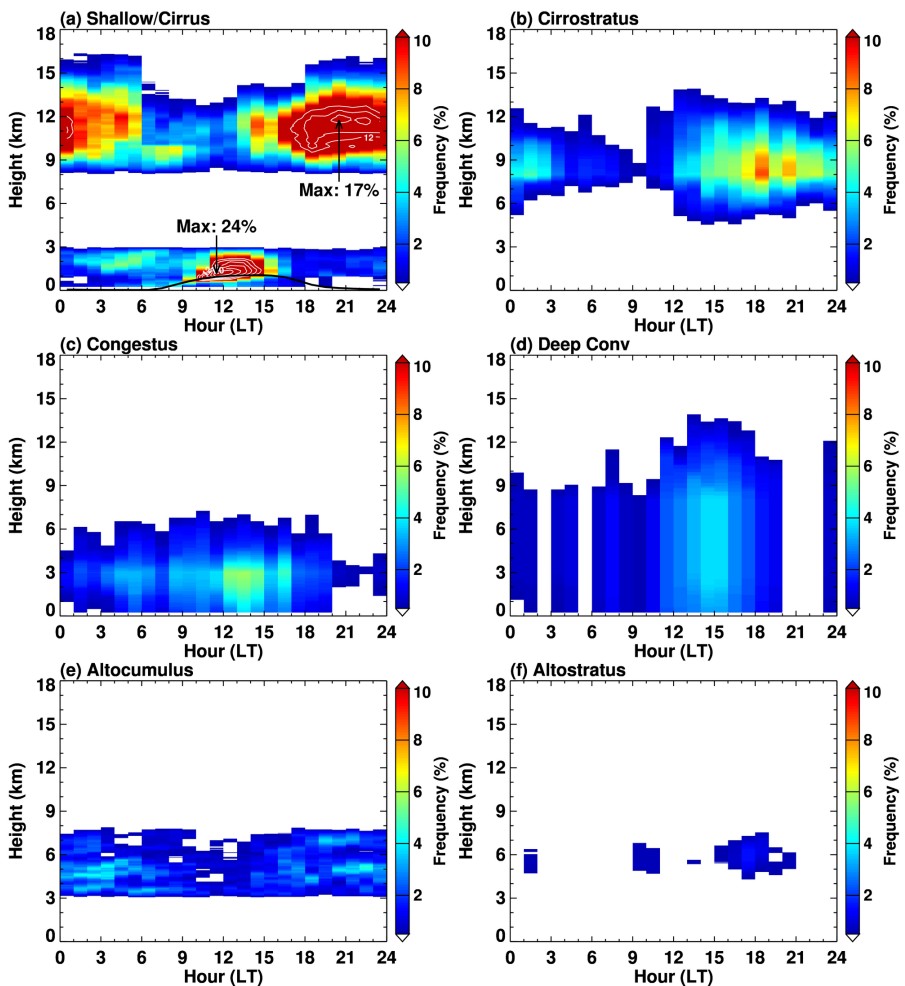

5 **Figure 9:** As in Fig. 6, but for dry season conditions.





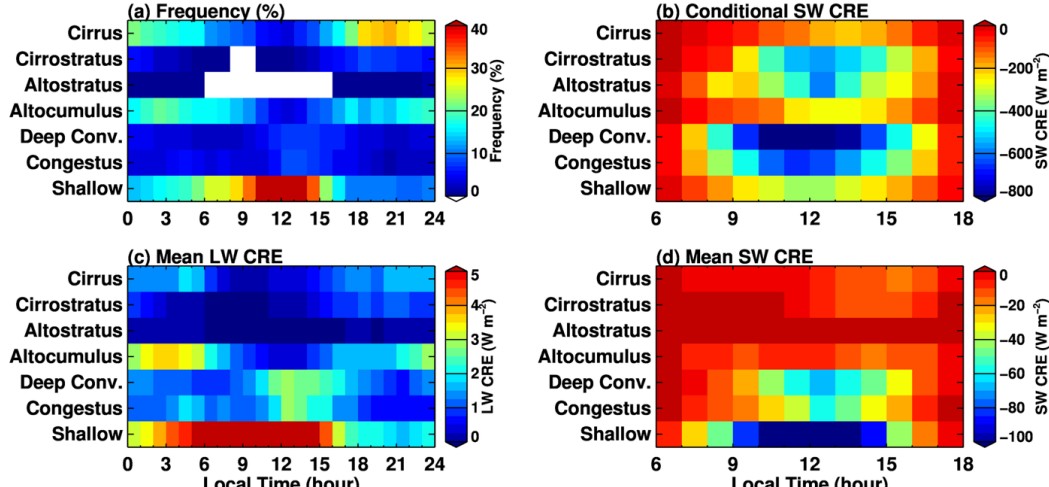

**Figure 10:** Mean cloud frequency and CRE for all GoAmazon2014/5 data. (a) Cloud frequency of occurrence as the lowest
cloud in the column as a function of the diurnal cycle (x axis), (b) conditional SW CRE, (c) mean LW CRE (frequency of
occurrence times the conditional LW CRE), and (d) mean SW CRE (frequency of occurrence times the conditional SW
CRE). Note (a,c) are for all hours and (b,d) are for daytime hours only. The white boxes are hours with insufficient data.



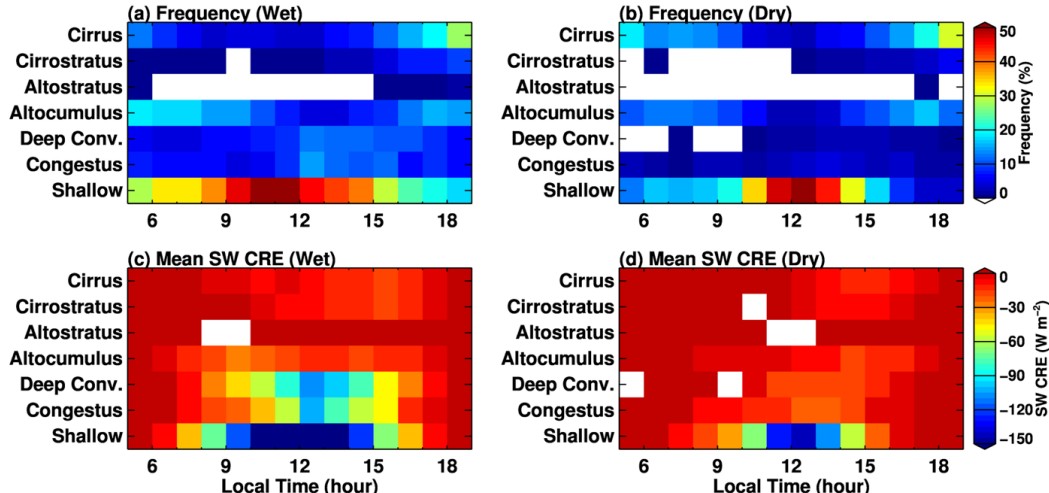

5   **Figure 11:** Wet (left column) and dry (right column) season comparisons for cloud frequency and SW CRE. (a,b) Cloud

frequency of occurrence as the lowest cloud in the column, as a function of the diurnal cycle (x axis). (c,d) Mean SW CRE

(frequency of occurrence times the conditional SW CRE). The white boxes represent hours with insufficient data.





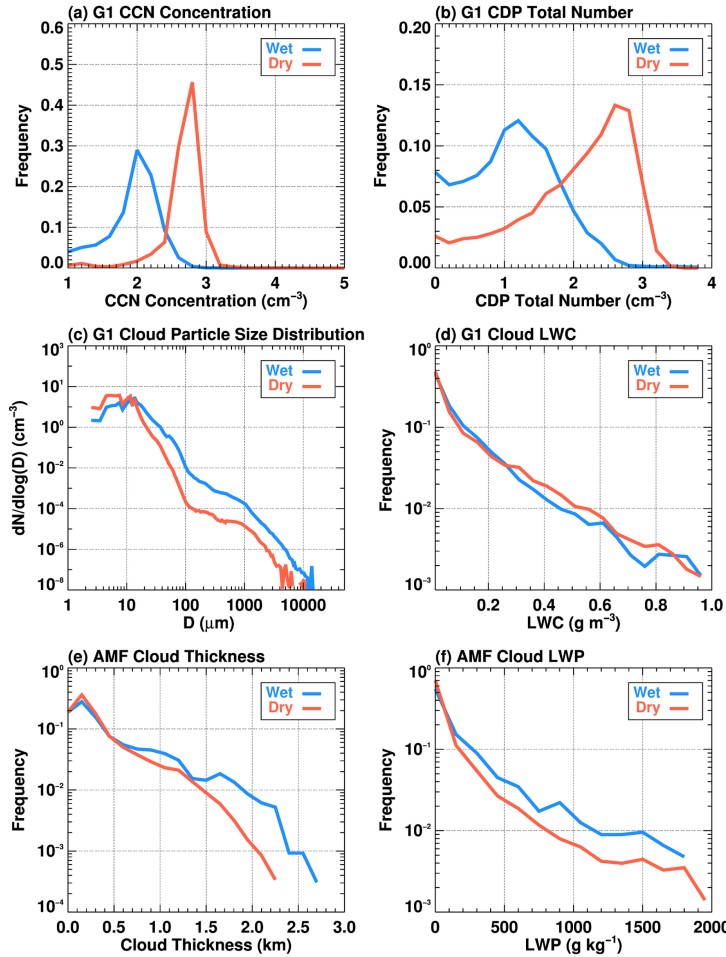

**Figure 12:** Shallow cumulus cloud micro- and macro-physics observed by AAF G1 aircraft and AMF surface

5 instrumentation at the T3 site during the two IOPs in GoAmazon2014/5. (a) Cloud condensation nuclei number

concentration, (b) cloud droplet total number concentration, (c) cloud particle size distribution, (d) cloud liquid water content

(LWC), (e) cloud thickness, (f) cloud liquid water path (LWP).