# Peer review of "Cloud Characteristics, Thermodynamic Controls and Radiative Impacts During the Observations and Modeling of the Green Ocean Amazon (GoAmazon2014/5) Experiment"

_Atmospheric Chemistry and Physics, 2017_

## Referee Comment (RC1) · Anonymous Referee #1 · 21 Jun 2017

This manuscript summarizes observational data collected during the GoAmazon campaign, emphasizing on cloud/precipitation-related properties, the dry-vs-wet season contrast, as well as diurnal variation.

As the first field campaign in tropical rainforest region so comprehensive, I think the results presented here are interesting, and will be useful to the community. Overall the manuscript is well-written, and I look forward to seeing this manuscript published on ACP. I do have four general comments that the authors could consider, and would, in my opinion, make the manuscript more accessible and useful to readers, especially

those from modeling side.

General comments:

First, proxies of uncertainty/variability (e.g., standard error, standard variation, or interquartile range) could be added to Figure 7 and Table 2 (and Table 3 if the authors have the necessary data). While Table 2 has been visualized, with some variation/extension, in Figures 10 and 11, an additional figure with values listed in Table 2 plus uncertainty/variability could make the results easier to digest.

Second, some analysis of TRMM 3B42 data have shown that the diurnal cycle at/around the GoAmazon site is rather unique compared with other tropical locations over land because the precipitation at the site peaks around noon instead of late afternoon (e.g., this figure from the AMWG Variability Diagnostics website: http://www.cgd.ucar.edu/cms/cchen/VDIAG_cam5_F2000_var/hourly/obs/TRIMM_PRECT.50N50S.dc.ANN.png), and Hourdin et al. 2013 (LMDZ5B: the atmospheric component of the IPSL climate model with revisited parameterizations for clouds and convection. Clim. Dyn. Figure 17, lower panel) seems to be able to reproduce this intriguing pattern of diurnal cycle with their modified convective scheme (of course, with a caveat, i.e., how the diurnal cycle is defined). This characteristic of precipitation diurnal cycle is somewhat consistent with Figure 7 in Tang et al. 2016 (ACP) or Figure 7 in the present manuscript. I think it would be helpful if the authors could, to the best of their current knowledge, and with the dry-vs-wet season contrast at the GoAmazon site in mind, add a short paragraph or two to comment on the uniqueness of the GoAmazon site (i.e., whether the diurnal cycle at this site is really different from other sites) and, if this site is indeed unique, to synthesize the similarities and differences of the precipitation/CF diurnal cycle at the GoAmazon site compared with other tropical locations over land (e.g., the larger Amazon basin, and/or Congo basin and maritime continent).

Third, partitioning precipitation (e.g., into convective vs stratiform precipitation) can be tricky, and the way adopted by different parameterizations differ. In some cases,

it may be difficult (if not impossible) to draw an analogy between observation and a parameterization. Therefore, it would be helpful if the authors could add a line or two to give some details regarding the definition of convective and stratiform rain (I thought the former is defined as all precipitation associated with shallow, congestus, deep convective, and altocumulus, but this would be inconsistent with the description in p11, l. 3: '... stratiform precipitation (identified as "Deep Convection" in ... ' which is not obvious to me since bright band is not used as a criterion for the could-type algorithm ...). Furthermore, given the potential issue of precipitation partitioning, precipitation rate conditioned on different cloud types could eventually be even more valuable for modeling groups (e.g., add an additional figure like Figure 11, but for precipitation rate; this last suggestion is totally optional for the authors).

Fourth, both Figures 10 and 11 seem to put more emphasis on SW than LW. Both SW and LW are important for quantifying local energy budget, and some recent modeling studies have demonstrated that LW could impact the development and maintenance of organized system, which has been documented by Bu et al. 2017 (The influences of boundary layer mixing and cloud-radiative forcing on tropical cyclone size. JAS), and the LW feedback is essential to convective self-aggregation under certain conditions, which has been summarized in Wing et al. 2017 (Convective self-aggregation in numerical simulations: a review. Surv. Geophys. Sec. 3.2). Given this increasing interests in LW-related processes, the authors could consider to add additional LW information to Figures 10 and 11.

Specific comments:

p3, l. 13: "...unique precipitation cycles as compared to the conditions over the larger Amazon basin'. Please see my second general comment.

p3, l. 24: "environmental forcing datasets" could be more specific.

p4, l. 7: "a cloud-type classification algorithm" refers to Table 1?

p4, l. 27: "Figure 2 . . . average daily profile . . . " it would be nice if the original temporal frequencies of the raw data can be mentioned here (from p. 8, l. 17, hourly profile estimates are used).

p5, l. 2: "CAPE, CIN and . . . and heightened moisture." Figure 3 (not 2) shows that CIN behaves differently in wet vs dry seasons, but not so much for CAPE. But in any event, the behavior of CAPE and CIN described here is not clear from Figure 2.

p5, l. 22: "increased CIN" in terms of magnitude, regardless of its sign?

p5, l. 26: "total advection. . . of moisture". I assume that this means $-\boldsymbol{u} \cdot \nabla q$ instead of $\boldsymbol{u} \cdot \nabla q$.

p5, l. 33: "The evening and early morning hours exhibit upward air motion confined below 3-4 km". How should we interpret this feature? For instance, based on the first law of thermodynamics, we have $\omega = \dot{p} = p\dot{Q}/c_v T - \gamma p \nabla \cdot u$, where $\dot{Q}$ represents the diabatic heating/cooling rate. Can wc explain the feature by low-level radiative cooling?

p6, l. 2: "positive advection of moisture" means $-\boldsymbol{u} \cdot \nabla q > 0$? (and p9, l. 8.)

p6, l. 2: "Between 4-8 km . . . (Fig. 4g)." This is not clear from the figure.

p7, l. 2: "The ARM 95-GHz W-band ARM . . ." one of the two ARMs seem redundant.

p7, l. 32: "highlighting locations . . . ARSCL methods." It seems to me that the determination of cloud-top is improved by RWP, and cloud-base by MPL. Is this summary correct?

p8, l. 16: ". . . Table 2." Please see my first comment. It would be helpful to mention at this point that a variation of Table 2 is plotted in Figures 10 and 11.

p8, l. 18: "Measurable precipitation (> 1 mm) . . . light/trace precipitation (< 1 mm)." Is this accumulated precipitation over one day or one hour?

p8, l. 24: ". . . below normal . . ." A number representing the "normal" accumulated

precipitation could be helpful.

p8, l. 27: "... convective and stratiform ..." Please see my third general comment.

p9, l. 30: "inspection of large-scale ... thermodynamics." Totally understandable statement, but isn't it true that the diurnal cycle of high-level clouds is tied to the diurnal cycle of the large-scale dynamics and thermodynamics through deep convection?

p10, l. 13: "profile methods also distinguish columns with convective vertical air motions...as 'convective'." Please see my third general comment. And further details regarding the profiler methods could be informative.

p10, l. 24: "the difference in the mean rainfall rate are less pronounced, implying dry season convection as stronger (instantaneously), since the convective cell coverage is also reduced during the dry season..." Recently Schiro 2017 (Thermodynamic Controls on Deep Convection in the Tropics: Observations and Applications to Modeling) examined animated radar data for the GoAmazon campaign period, and reached a similar conclusion.

p10, l. 28: "... organized systems pass over T3 primarily in the morning hours during the wet season..." Is this consistent with Figure 8(d)?

p10, l. 32: "... with the dry season having less organized cloud contributions ..." Will Figure 9 be similar to Figure 8 if both figures are composed for only precipitating days?

p11, l. 3: "... stratiform ..." Please see my third general comment.

p11, l. 6: "... consistent with a response to increased surface heating and an increase in the surface latent heat flux..." The increments are defined with respect to certain reference values, but it is not clear what these reference values are.

p11, l. 13: "This pattern suggests ... under wet season conditions." If the shielding issue mentioned right after this statement is real, the pattern is also due likely to the suppressed contrast of cirrus cloud, isn't it?

[Figure]

p12, l. 22: "... sample sizes are potentially too small ..." Please see my first general comment. Including uncertainty/variability is a potential solution.

p12, l. 31: "... precipitating convective clouds..." Please see my third general comment. The definition of precipitating and non-precipitating clouds, i.e., the threshold, is not clear.

p16, l. 15: "... suggest clouds influenced by aerosol tend to have larger concentration of smaller droplets and fewer precipitation sized drops for clouds with similar LWC." With the evidence presented in the manuscript (no information about chemical composition, though seasonal wind direction might implicate), it is unclear that how the wet-vs-dry season contrast for larger-scale environment dynamic/thermodynamic conditions would affect the cloud-microphysical processes (e.g., collision-coalescence, precipitation scavenging, ... I suspect that this question can only be answered by later modeling studies), I would modify this sentence as something like "... is consistent with the hypothesis that clouds influenced by aerosol tend to have larger concentration of smaller droplets and fewer precipitation sized drops for clouds with similar LWC."

Figure 2(a), I suppose the hydrometeor frequency is defined with respect to a threshold. Assuming this is true, it would be nice to know the threshold.

Figure 2(c), two additional horizontal lines could help the readers better capture the evolution of CAPE and CIN.

Figure 5(d), the rain rate is discretized with non-trivial units.

Figure 7(b), it is not clear in the caption how a precipitating day is defined by the condition > 1 mm/hr. Does it mean the daily accumulated precipitation > 24 mm, or hourly accumulated precipitation > 1 mm for any hour of the day (also note p12, l. 28)?

Figure 7(c), it is not clear that whether the fractions are calculated for all days or for only precipitating days.

Figure 10(a), the frequency of altostratus is 0 from 6 to 16 (assuming my reading of

the color bar is accurate). Following the usual definition of conditional average, the frequency should be in the denominator. If this is how conditional SW CRE is defined, how could it be well defined for the same period when frequency is 0?

Figure 12(f), The units of LWP are g/kg, which is different from the conventional definition and Figure 5(d). A typo maybe?

---

## Referee Comment (RC2) · Anonymous Referee #2 · 27 Jun 2017

Overview

This manuscript presents a nice overview of unique datasets from the central Amazon, specifically focusing on vertical profile measurements of thermodynamic conditions and cloud boundaries with collocated surface radiative flux observations. It doesn't get into much detail on any one science question, but that is okay for an overview paper, especially since these datasets are not discussed in detail in Martin et al. (2016, 2017) overview papers. The results showing differences between wet and dry seasons are expected from previous studies, and the most interesting new results are the estimated

cloud radiative effects of different cloud types as a function of season and diurnal cycle showing mean properties over a year that are similar to shortwave and longwave cloud radiative effects at a very rainy site in the Tropical Western Pacific. Another interesting finding is that cumulus clouds tend to be deeper with greater liquid water paths in the wet season than the dry season despite similar liquid water contents in the clouds and larger droplets in the wet season clouds. My primary concern is that the authors don't delve far enough into these interesting findings leaving the paper very descriptive with open questions. Being an overview paper of newly available datasets, I don't expect as much scientific investigation as in other types of papers, but I think that there are fairly simple and straightforward ways that the authors could attempt to explore some of the reasons for the aforementioned interesting results, as described further in major comments 5 and 6.

Major Comments

1. There are a lot of coauthors on this paper, and I can't help but wonder how each coauthor contributed to the paper. Does each coauthor meet ACP's guideline that "only persons who have significantly contributed to the research and paper preparation should be listed as authors"? What did each coauthor contribute?

2. The CAPE values seem extremely high. For example, they are far higher than those from Manaus in Machado et al. (2004) and in Williams et al. (2002) in Rondonia, Brazil (both papers that you cite). This could be a result of having mid-day soundings, but have you checked to make sure that these are correct? Was any comparison made with the 12 and 00Z Manaus operational soundings? And what is causing the CAPE to rise so dramatically to near 4000 most days? This is an interesting feature. Is this confined to lifted parcels within a thin PBL layer, so that mixing erases much of it and prevents extreme convection from occurring?

3. Increase the text size on Figure 3. It is currently unreadable. Also, the units in Figure 12a-b are off by 2 orders of magnitude.

4. The hour to hour noisiness of rain rate in Figure 7 seems to indicate modest sample size, perhaps not surprising for a point location, even if observed for nearly 2 years. Does this impact the robustness of conclusions that require sub-setting of the data (e.g., diurnal cycle, seasonality)? The peak in rainfall at noon is also sooner than is typically observed in most land locations, and it doesn't seem to match up with the peak congestus and deep convection cloud fractions in Figures 6, 8, and 9. Is there a reason for this?

5. The similar CRE between Manaus and Manus is interesting, and perhaps expected, especially relative to Darwin and Nauru, which are not typical of many rainy, tropical places because of the dry air aloft that often impacts them. However, breaking down the overall SW and LW CRE at each site by frequency of cloud type and conditional CREs for each cloud type like in Table 2 would be much more interesting. Are some cloud types similar while others are different between sites? How do these similarities and differences impact the overall SW and LW CRE? Are there differences in the diurnal cycle of cloud type frequencies and CREs? Ideally, it would be extremely informative to investigate potential thermodynamic relationships with cloud frequency or CRE at the different sites or to pick a key cloud type like shallow cumulus and relate its CRE with its depth and LWP at each site. I'm not suggesting that I expect you to do all of this, but it seems that it would be fairly straightforward and not very time consuming to investigate a little more, placing this new dataset into better context with other well-observed sites.

6. Given the datasets available that are highlighted in this paper, it seems like it should be straightforward to investigate the cause of deeper cumulus clouds with greater LWP in the wet season as compared to the dry season, but with similar LWC. Is the cumulus cloud depth related to the CIN, relative humidity just above the boundary layer, both? When cumulus cloud depth is controlled for, do the dry season clouds have a stronger SW CRE caused by greater numbers of smaller droplets? How much of an increase in cloud depth is required to offset the cloud droplet SW CRE? These are questions that could probably be examined without too much effort that are important

to understanding how all of these observations connect together through processes.

7. ACP requires a "Data Availability" section on how all research data can be accessed, which goes before the "Acknowledgements" section. Please insert this section.

Minor Comments

1. On page 5, lines 5-7, I have trouble seeing higher CAPE and lower CIN in the transition periods in Figure 2. It looks fairly constant and the CIN appears lowest in the wet season and highest in the dry season, consistent with Figure 3, so can you clarify this sentence or better show it in the figure? It is then stated on page 11, lines 23-24 that wet season conditions favor weaker CAPE and dry season stronger CAPE, but the figure don't seem to show that. There is very little difference in Figure 3a, with enough spread that the differences between wet and dry season CAPE is likely not statistically significant.

2. On page 5, lines 16-17, it is stated that CIN decreasing during the day with CAPE peaking at mid-day is consistent with development of convection breaking the capping inversion and consuming CAPE, but I don't understand this argument. Isn't the layer of CIN (not necessarily an inversion, by the way) simply reduced through boundary layer mixing induced by daytime heating, while the same daytime heating warms the boundary layer and increases CAPE?

3. Is the moisture advection in Figure 4 the same as moisture convergence? How is it calculated? It is difficult to interpret 3-D advection. How much of the advection in Figure 4 is a result of the vertical component? Can it be stated if vertical mixing is the dominant component?

4. On page 7, it is stated that the RWP is used to reconstruct cloud boundaries up to 13 km, but really, isn't the RWP observing precipitation boundaries due to its sensitivity limitations? And because of these same limitations, doesn't it underestimate cloud top? These seems apparent in Figure 5, for example. If so, I recommend mentioning

this limitation.

5. The CRE estimates depend on accurate estimates of the clear sky radiative fluxes. What causes the sharp decrease in clear sky longwave flux at 15Z in Figure 5f that appears coincident with the edge of the deep convective system? Is this accurate? If not, is there a bias in some situations that impacts longwave CRE?

6. It is mentioned that low level clouds impact the detection of upper level clouds by the micropulse lidar and ceilometer. Is this what caused the discontinuities in cirrus cloud fraction at 6 and 18Z in Figure 6a? If so, can this be stated? And does this mean that cirrus is significantly underestimated at the uppermost levels?

7. On page 9, there are several places where the wording doesn't seem accurate:

a) On lines 5-6, it is stated that shallow clouds dominate the early morning hours, but to me, it looks like they continue well into the afternoon.

b) On line 12, it is stated that congestus and deep convective clouds are prominent from mid-late afternoon, but it looks like they are prominent starting right at noon, and peak cloud fraction looks to be between 12 and 1 PM for congestus and 12 and 2 PM for deep convection rather than at noon, as is stated in the paper.

c) On line 21, it is stated that the convection aligns with the mid-upper level vertical motion, but there is a secondary maximum in deep convection overnight when there is descending mid-upper level motion, so these don't seem perfectly aligned to me.

d) On line 24, the pre-dawn peak in altocumulus is its primary peak, not its secondary peak.

e) Lines 25-27: The wording here is confusing. What does precipitation have to do with congestus cloud fraction peaks? Please clarify.

8. Stating that 103 wet season days produce 1600 mm of rainfall and 52 dry season days produce 600 mm of rainfall doesn't necessarily lead to a factor of 2 difference in

average rain rate. I can see that differences in Figure 7, but the argument from the number of days and precipitation perspective requires knowledge of how many total wet season and dry season days were sampled.

9. The statements on page 10, lines 5-7 and lines 23-25 seem contradictory with one saying that relative convective intensity is not much different between wet and dry seasons, while the other says that dry season convection is stronger. I personally don't see evidence in Figure 7 that dry season rain rates are more intense with conditional rain rates in both seasons that are similar.

10. On page 10, line 28, how are you defining "organized systems"? And similarly, on page 11, line 1, how is "organized cloud" defined?

11. It's not surprising that clouds and thermodynamics are not strongly correlated when viewed from a stationary vertically pointing perspective, but this shouldn't be confused with them not being correlated over a larger scale from a Langrangian perspective. When you state that weak correlations are found between cloud behaviors (isn't cloud state a better term here than behavior?) and thermodynamic parameters, what time scale is this correlation being computed on and is it a time lag correlation?

12. On page 12, line 10, I believe you can delete "Compared to SW CRE" to make the sentence read more clearly. This sentence mentions SW CRE being much larger than LW CRE, but SW CRE is 0 at night and the surface energy balance is much different at night than during the day, so a given LW CRE at night, even if small compared to the larger daytime SW CRE, could have just as significant of impacts on variables such as surface temperature, couldn't it?

13. On page 13, line 2, insert "and" between "(SW)" and "longwave" with a comma after "fluxes". And on line 29, add an "s" onto "peak".

14. On page 15, line 15, it is stated that sharper CAPE and CIN contrasts during the wet season and transitional periods enhance the likelihood for deep convection to have

organized components. First, how are organized components defined? Second, how is this known? I didn't see any evidence presented that shows this and important components of mesoscale convective system growth such as vertical wind shear were not discussed. Mid-upper level humidity and vertical motion are also potentially important factors aside from CAPE or CIN.

15. Consider rewording "supports local congestus to deeper cloud triggering" on page 15, line 19.

16. On page 16, line 1, consider rewording to "between maritime-like 'active monsoon' conditions with widespread clouds and precipitation and continental-like 'break monsoon' conditions with less clouds and precipitation, but more intense deep convection".

17. On page 16, line 10, insert "cumulus" after "thicker".

18. What is meant by "natural cloud laboratory" on page 16, line 14?

19. The last paragraph on page 16 has some confusing wording in spots. For example, the third sentence seems out of place in the paper since the Manaus plume was not discussed at all in this study. Additionally, the last two sentences don't seem worded well. Can you attempt to clarify what is meant in these sentences?

---

## Referee Comment (RC3) · Anonymous Referee #3 · 27 Jun 2017

Review of "Cloud Characteristics, Thermodynamic Controls and Radiative Impacts During the Observations and Modeling of the Green Ocean Amazon (GoAmazon2014/5) Experiment" by Scott E. Giangrande et al.

Overview

This manuscript describe a unique long term set of observations, as well as aircraft IOPs, in the Amazon. These measurements are necessary to help better constrain global climate models and parameterizations of clouds and precipitation for regions,

like the Amazon, that have been challenging to simulate. Due to the long term nature of the observations they are able to look at cloud and precipitation over the diurnal and seasonal time scales. Their observations of clouds, are limited from these long term sites due to the nature of their 1D observations. From an aircraft based background or a satellite perspective, these estimates of cloud fraction aren't ideal. The works isn't groundbreaking, but it is a good overview paper of the observations from the project that aren't addressed in the other GoAmazon papers.

Main Comments

1. How often are multi-layer clouds observed? The authors discuss the fact that if they removed the multi-level cases they wouldn't have enough data for analysis. This is a concern when thinking about sorting data by convective and stratiform. Are these categories meaningful if it's only based on the lower most data? What if there are both convective (low level) clouds and higher level sratiform clouds that are obscured? Does this have an impact on the rain rate? Are the rain rates difference from convective clouds only vs. multi-level louds?

2. Estimates of uncertainty are missing and should be addressed. Also, on several figures an idea of the sample size would help put the data into context. It is difficult to evaluate the data when it is unclear how much data is actually included in the figures.

3. I'm confused why they bring in the other locations (Darwin, Nauru and Manus)? These data are not discussed anywhere else in the paper except in section 4 and to add a paragraph in section 5. It seems out of place, it could be part of another paper.

4. Have the authors considered using MODIS CF to get an idea how well their CF estimates match satellite observations? They can also use CALIOP to see how their estimates of multi-level cloud classifications compare to the ground measurements. Finally, they can use TRMM or GPCP data to get regional estimates of precipitation. These would put their work into a larger scale context.

**Minor Comments** There are often generalizations and wording issues that make their points less clear. These issues can be fixed by modifying their text. See specific comments below:

Page 2 Line 30 – "cloud study complement to the..." → "cloud study to complement the..."

Page 3 Line 3 – "cloud types and contrasts" what do you mean by "contrasts", do you mean differences in atmospheric conditions and thermodynamics, seasonal or diurnal changes. This is vague.

Page 3 Line 4-5 – Wording: "This analysis includes additional relationships to campaign aircraft in-cloud observations when available." → "This analysis includes additional relationships between in-cloud aircraft campaign observations when available."

Page 3 Line 10-11 – What does "possibly maritime-like atmospheric conditions" mean, this is a vague comment and needs clarification.

Page 3 Line 12 – Clarify what you mean by a region of "underlying moisture." How would you define this, humidity, PW?

Page 3 Line 16 – "work has found a robust relation..." → "work has found a robust relationship..."

Page 3 Line 19 – Clarify "cloud lifecycle complexity" what complexities are you suggesting are additional versus not-additional, the wording here is unclear.

Page 3 Line 34 – What are the "environmental forcing data sets?"

Page 4 Line 8 – The pencil-beam/soda-straw description is not necessary.

Page 4 Line 9-10 – Please elaborate on how CF is described, so 50% cloud cover is recorded when there are cloud present for 30 minutes out of an hour? This seems strange compared to thinking of CF as a fraction of an area if I am interpreting the description correctly.

Page 4 Lines 17-18 – Does this choice, of using the maximum virtual potential temperature, result in a bias towards higher CAPE? Did you try a max, min and mean virtual potential temperatures to see how this changed the results?

Page 4 Lines 32 – wording is strange for "on their suitability" I suggest rewording this sentence.

Page 4 Line 32 – Page 5 Line 1 – What is shown in panel a) of Figure 2. The text here suggests cloud fraction, but the label says "hydrometeor frequency." Clarify.

Page 5 Line 7-8 – Referring to Figure 3, it is not obvious in the CAPE panel that there is a difference between the wet and dry periods. They look the same based on Figure 3a.

Page 6 Lines 27-28 – How are the SCMWF analysis outputs constrained using the surface rainfall? The constraints are not identified in the text. Is it precip or no-precip? A certain amount of precip with a specific threshold? Clarify.

Page 7 Lines 31-32 – (also noted in the figure comments) – where is the red bars located? Are they the thin lines above panel c? If so, a new way to note this should be found, it is not clear or easy to use these bars for the purposes described in the text.

Page 8 Lines 5-6 – Why are there no stratus or stratocumulus categories? The only stratiform clouds are altocumulus and cirrocumulus? Are there just so few of these cloud categories that you are leaving them out?

Page 8 Lines 24-25 – You bring up that the 2014-2015 rainy season maybe be different than climatology. Perhaps it would be beneficial to show a monthly climatology for a long period of time to which you can compare the 2014-2015 rainy season? This way the readers can know how different this particular year is from the climatological average. After reading this comment I was left wondering if these results are just a special case or if they are in fact representative of this region in a general sense.

Page 8 Lines 25-27 – This would be a good place to include a more detailed description

of how you calculate the precipitation as a function of convective and stratiform clouds. This is difficult and the method isn't clearly stated.

Page 9 Line 19 – What do you mean by "rare?" How often do these sea breeze intrusions occur? How many times did this happen over the study period? What impact do these intrusions have on the results?

Page 10 Line 8 – Please describe how you separate the rainfall rates in to convective and stratiform types. Is this based solely on the cloud classification (as shown in Figure 5 c)?

Page 11 Line 3 – "stratiform precipitation (identified as "Deep Convection" …" this is confusing. Do you mean stratiform as in cirrostratus? Usually when I see the term stratiform I think of stratus or stratocumulus. Clarify this section.

Page 12 Line 14 – The "green ocean" comment again, it's not really fully discussed in the beginning section (Page 3 Line 10) where it is first mentioned so it's strange to mention it again.

Page 12 Line 28 – "better" this is not a descriptive term. What makes it better? Be more specific.

Table Comments

Table 1-3 – The format of these tables is difficult to read in the current format. I'm assuming that they will be formatted differently when published.

Table 3 – Why are the authors including these other locations that are very far from the Amazon. When reading the text, there doesn't seem to be a solid justification for this other than to make a quick comparison.

Figure Comments

Figure 2 – Continue the IOP dashed lines all the way to the top of the figure through panels a, b, and c. Perhaps make solid lines to section off the wet and dry periods as

well. This would be helpful for knowing where the cut offs are. I drew them on myself to make it easier for me to see when they started and ended. What is "hydrometeor frequency" shown in panel a). In the test it suggests CF (Page 4-5 Lines 32 -33 "The more pronounced shifts during the wet season include increased CF in the mid0to-upper troposphere (between 3-10 lm, Fig 2a). So is it CF or something else that is being shown as frequency. Please clarify.

Figure 3 – Axes labels are small and hard to read. What is going on with the 19 am data for CIN. Why are they essentially a point?

Figure 4 – Hard to read the panel labels for (g), (h), and (i). Perhaps move them outside and next to the left side of the figures for all the panels.

Figure 5 – It is difficult to read the label or panel c), it is obscured by the cirrus clouds. The green and red line above panel c), is that the red bars referred to on Page 7 Lines 31-32. If so, these are near impossible to see clearly.

Figure 6 – The white contours (starting at 10%) are difficult to see, perhaps increase the line thickness.

Figure 7 – For a) How many days are included (are there equal number of days in all the time bins?) b) same as a), how much data is included? c) Define the fractional accumulation more clearly (convective/stratiform). Does the number of samples change for each time bin? Are some time bins 100 samples (80 wet 20 dry) while others are 10 samples (8 wet 2 dry). It would be nice to know how much data is going into theses curves.

Figure 8 & Figure 9 – Perhaps these two can be merged so you can easily compare the difference between the wet and dry seasons. A difference panel, or just showing the difference between wet and dry would be a good way to show were the differences are most pronounced. As with Figure 6, the white contours (starting at 10%) are difficult to see, perhaps increase the line thickness.

[Figure]

---

## Author Comment (AC1) · 30 Aug 2017

The comment was uploaded in the form of a supplement:
https://www.atmos-chem-phys-discuss.net/acp-2017-452/acp-2017-452-AC1-supplement.pdf

---

## Author Comment (AC2) · 30 Aug 2017

**Response to Anonymous Referees,**

*"Cloud Characteristics, Thermodynamic Controls and Radiative Impacts During the Observations and Modeling of the Green Ocean Amazon (GoAmazon2014/5) Experiment"*

**Scott E. Giangrande et al.**

The authors would like to thank all reviewers for their helpful comments and suggestions. We have responded to all reviewers in a single document, since several comments are similar. As a brief summary, the revisions to the manuscript include the following highlights:

- We have modified several of the previous images (fonts, lines, sizing, etc.) and added a few new / revised plots that combine image ideas.
- The manuscript has incorporated several changes in response to reviewer comments on uncertainty/representativeness. Specifically, we have included a new image and discussion section (in Section 3) in response to the representativeness of these observations as compared to satellite-based platform references (MODIS, GOES) – Reviewer 3; This also is one practical way to approach some of the observational uncertainty concerns from Reviewer 1.
- We have attempted to clarify the text on the convective/stratiform precipitation and how this regime segregation is related to the cloud classification – We note, this separation is different than 'deep convective' cloud regime classifications, in that these deep convective regimes may contain convective and stratiform precipitation components.
- We have included a 'Data Availability' section, as required by the journal.

The individual reviewer comments and responses are included in the following document (author comments in **bold**, reviewer comments in *italics*).

**Response to Anonymous Referee #1**

*This manuscript summarizes observational data collected during the GoAmazon campaign, emphasizing on cloud/precipitation-related properties, the dry-vs-wet season contrast, as well as diurnal variation. As the first field campaign in tropical rainforest region so comprehensive, I think the results presented here are interesting, and will be useful to the community. Overall the manuscript is well-written, and I look forward to seeing this manuscript published on ACP. I do have four general comments that the authors could consider, and would, in my opinion, make the manuscript more accessible and useful to readers, especially those from modeling side.*

**We thank the reviewer for their kind words and we hope our revisions are sufficient to address many concerns of this reviewer.**

*General comments:*
*First, proxies of uncertainty/variability (e.g., standard error, standard variation, or interquartile range) could be added to Figure 7 and Table 2 (and Table 3 if the authors have the necessary data). While Table 2 has been visualized, with some variation/extension, in Figures 10 and 11, an additional figure with values listed in Table 2 plus uncertainty/variability could make the results easier to digest.*

**This is a very good comment, but difficult to answer. These observations are challenging to bound as they merge several ARM observations to maximize designation of clouds. Another concern when addressing this comment is that in some instances, the suggested 'proxy' ways to report spread (standard deviation) may add confusion (e.g., it is not trivial to explain what those values represent).**

**To begin, there are not many independent observations to act as reference for cloud frequency (esp. under all cloud conditions reported). In our response to Reviewer 3, we added to Section 3 a comparison with independent passive satellite observations (MODIS, GOES) for Shallow Cumulus; Note, even this approach implies new complexities (interpretation) because those cloud sampling techniques are different. Yet, this is one way to address sampling/representativeness concerns, as well as to gauge the uncertainty of T3 cloud observations (existing reference).**

**Second, given the large diurnal variability and the nature of the instantaneous observations, the authors suggest it may not always make sense to condense what is reported by Tables 2 or 3 (or Figure 7) down to a single value (e.g., reporting a single standard deviation). One example for this reviewer is specific to Figure 7, but a case where we believe the application falls short of the anticipated outcome. Below, we plot the standard deviations for Figure 7ab, noting that these values are substantially larger than the mean rainfall rates (would require also a significant re-plotting). Now, this is not a surprising result, given that these are 'instantaneous' (5-minute) rainfall rate measurements as compared to a mean value for a given hourly interval:**

[Figure]

Here, it would not be difficult to include these references to our revised manuscript; However, we question the value for presenting this as 'uncertainty', insomuch that the instantaneous values used to estimate those standard deviations also reflect the many contributions from zero value data points, etc.; Rather, we have included the total accumulation plot (new Figure 7c), which should help empress that the number/probability for observing surface rainfall during certain hours are substantially lower.

Similarly, the variability of cloud frequency and CREs reported in Table 2 are difficult to be condensed into a single number. Because of the nature of a point measurement, occurrence of cloud at any given instance is either 1 or 0, the standard deviation for the same hour across different days are oftentimes as large as, or exceed, the mean value (see figure below). This is true for most of the cloud types except shallow cumulus during the middle of the day. In addition, the standard deviations themselves have a large diurnal variability. Therefore, the authors believe it does not make sense to report a single standard deviation (or similar quantity representing variability) number for each of the cloud type in Table 2. The only variable that does not have a strong diurnal variability is SW Transmissivity.

Thus, to respond to the Reviewer we added the standard deviations of SW transmissivity calculated across the diurnal cycle to Table 2.

[Figure]

Mean (left column) and standard deviation (right column) of cloud frequency of occurrence as the lowest cloud in the column for (a,b) all data, (c,d) wet season, (e,f) dry season.

In the case of Table 3 and its associated discussion, we agree that more information could be valuable in this application. In this case, we have included a new figure into the revised Section 4 that helps demonstrate the full diurnal cycle of variables in Table 3, as shown below. Here, the hourly standard deviation is represented by the shaded regions.

[Figure]

**New Figure added to complement Table 3.**

*Second, some analysis of TRMM 3B42 data have shown that the diurnal cycle at/around the GoAmazon site is rather unique compared with other tropical locations over land because the precipitation at the site peaks around noon instead of late afternoon (author snip to save space)… I think it would be helpful if the authors could, to the best of their current knowledge, and with the dry-vs-wet season contrast at the GoAmazon site in mind, add a short paragraph or two to comment on the uniqueness of the GoAmazon site (i.e., whether the diurnal cycle at this site is really different from other sites) and, if this site is indeed unique, to synthesize the similarities and differences of the precipitation/CF diurnal cycle at the GoAmazon site compared with other tropical locations over land (e.g., the larger Amazon basin, and/or Congo basin and maritime continent).*

**One important addition to this manuscript is the comparison against available satellite ShCu observations that may also help demonstrate spatial representativeness in a manner not previously covered by other authors.**

**This topic (representativeness) was the focus of a related GoAmazon2014/5 effort by some of our coauthors – Burleyson et al. 2016 (JAMC). Their primary finding was that the diurnal cycle of deep convection around the GoAmazon sites is a superposition of locally forced convection and the inland propagation of the previous day's sea-breeze front. Their climatology showed that deep convection begins around noon around the sites, and has a broad peak in the mid-afternoon over the T3 site. There is also some variability among the various GoAmazon2014/5 sites that is related to localized circulations generated by the river breezes.**

**More to the Reviewer's original comment, the diurnal cycle at the T3 site is generally well correlated with the regional behaviors to within a few degrees of the site; We note that the correlation drops off sharply at distances more than a few hundred kilometers, and this is particularly important during the wet season. We would encourage the reviewers (or those reading these response) to download that paper for more information.**

**Here, we also include the following references to the Burleyson et al. study that were in our original and revised versions of the manuscript:**

**P3, L17-20: "Seasonal thermodynamical shifts, as well as additional large-scale sea-breeze front type intrusions into the basin (e.g., Cohen et al. [1995], Alcântara et al. [2011]), promote additional cloud lifecycle complexity and diurnal cycle of precipitation variability (e.g., Burleyson et al. [2016], Saraiva et al. [2016])."**

**P9, L17-20: "The T3 location exhibits a pronounced diurnal cycle associated with deeper convection (as also in Saraiva et al. [2016]). This pronounced behavior is further representative of the fortuitous placement for the T3 AMF site, wherein daily cloud lifecycles also phase well with propagating sea breeze intrusions over this portion of the Amazon basin [Burleyson et al. 2016]."**

**P11, L1-5: "Overnight and/or pre-dawn deep convection (e.g., organized, continuation) and additional local congestus development are more common in the wet season. The distinct nighttime enhancement in stratiform precipitation (identified as "Deep Convection" in Figs. 6d, 8d and 9d) during the wet season is consistent with previous findings that propagating convective cloud systems contribute to the observed diurnal cycle of deep convection [e.g., Burleyson et al. 2016, Tang et al. 2016]."**

*Third, partitioning precipitation (e.g., into convective vs stratiform precipitation) can be tricky, and the way adopted by different parameterizations differ. In some cases, C2 it may be difficult (if not impossible) to draw an analogy between observation and a parameterization. Therefore, it would be helpful if the authors could add a line or two to give some details regarding the definition of convective and stratiform rain (I thought the former is defined as all precipitation associated with shallow, congestus, deep convective, and altocumulus, but this would be inconsistent with the description in p11, l. 3: '... stratiform precipitation (identified as "Deep Convection" in ... ' which is not obvious to me since bright band is not used as a criterion for the could-type algorithm...). Furthermore, given the potential issue of precipitation partitioning, precipitation rate conditioned on different cloud types could eventually be even more valuable for modeling groups (e.g., add an additional figure like Figure 11, but for precipitation rate; this last suggestion is totally optional for the authors).*

**We have attempted to improve discussion on this subject. We agree that it may be confusing to readers that a single category of 'deep convective' clouds is associated with 'convective' and 'stratiform' rain components.**

*Fourth, both Figures 10 and 11 seem to put more emphasis on SW than LW. Both SW and LW are important for quantifying local energy budget, and some recent modeling studies have demonstrated that LW could impact the development and maintenance of organized system, which has been documented by Bu et al. 2017 (The influences of boundary layer mixing and cloud-radiative forcing on tropical cyclone size. JAS), and the LW feedback is essential to convective self-aggregation under certain conditions, which has been summarized in Wing et al. 2017 (Convective self-aggregation in numerical simulations: a review. Surv. Geophys. Sec. 3.2). Given this increasing interests in LW-related processes, the authors could consider to add additional LW information to Figures 10 and 11.*

**The LW feedbacks that previous studies and this Reviewer discuss are related to the LW cloud radiative heating profile (contributions from cloud and water vapor) in the atmosphere. This is different from the definition used in our study for "surface cloud radiative effects" (e.g., this is the explanation why Fig. 10c LW CRE is an order of magnitude smaller than SW CRE).**

Specific comments:
p3, l. 13: "...unique precipitation cycles as compared to the conditions over the larger Amazon basin'. Please see my second general comment.

**As in response to the general comment.**

p3, l. 24: "environmental forcing datasets" could be more specific.

**Revised to "*Large-scale forcing datasets (including advective tendencies and vertical velocities) over this region were also supported by ...*"**

p4, l. 7: "a cloud-type classification algorithm" refers to Table 1?

**Yes. This refers to the cloud classification algorithm performed on the combined RWP+ARSCL datasets that is partially summarized by Table 1, and for example also feeds into the calculations for Table 2, etc.**

p4, l. 27: "Figure 2...average daily profile..." it would be nice if the original temporal frequencies of the raw data can be mentioned here (from p. 8, l. 17, hourly profile estimates are used).

**It is unclear if this action is necessary / the authors are unsure how to respond - certain measurements in this figure are obtained at higher frequency than others (aka, between radiosondes versus radar, etc.), across the various plots. The ARM raw data sampling is more frequent than hourly for many instruments (typically seconds to minutes in the case of radar and/or gauge sampling), as noted in the associated instrument sections.**

p5, l. 2: "CAPE, CIN and ...and heightened moisture." Figure 3 (not 2) shows that CIN behaves differently in wet vs dry seasons, but not so much for CAPE. But in any event, the behavior of CAPE and CIN described here is not clear from Figure 2.

**Figure 2 shows the time series of CAPE and CIN. As also raised by other reviewers, we agree with this reviewer that it is difficult to view pronounced changes in these variables related to the separate wet and dry season intervals. We would agree this is easier to suggest these variations in terms of Figure 3 or dataset median values; For example, Figure 3 we believe does suggest that CAPE is higher during the dry season (especially at midday), and CIN is also greater (more negative) during the dry season. However, we have attempted to clarify what the ARM observations demonstrate in our manuscript (as compared to those observations of others).**

p5, l. 22: "increased CIN" in terms of magnitude, regardless of its sign?

**Agree. We now mention this refers to the absolute values of CIN.**

p5, l. 26: "total advection...of moisture". I assume that this means $-u \cdot \nabla q$ instead of $u \cdot \nabla q$.

**Yes. We revised the text to add the equations:**

**"Fig. 4 plots the diurnal cycle of the large-scale vertical motion (omega), total advection of moisture and relative humidity. The total advection of moisture is the sum of the horizontal and the vertical advections:**

$$q\_adv\_t = -V_h \cdot \nabla_h q - \omega \frac{\partial q}{\partial p} \text{ "}$$

p5, l. 33: "The evening and early morning hours exhibit upward air motion confined below 3-4 km". How should we interpret this feature? For instance, based on the first law of thermodynamics, we have $\omega = \dot{p} = pQ/c_vT - \gamma p \nabla \cdot u$, where Q represents the diabatic heating/cooling rate. Can wc explain the feature by low-level radiative cooling?

**We suggest that the vertical velocity feature can be explained by the low-level convergence and middle-level divergence in the evening and early morning (see the figure below). This may be related to the congestus clouds during this period. We have revised the text to add an interpretation of this feature:**

**_"The evening and early morning hours exhibit upward air motion confined below 3-4 km, due to the lower-level convergence and middle-level divergence (not shown), likely corresponding to the congestus clouds."_**

[Figure]

p6, l. 2: "positive advection of moisture" means $-u \cdot \nabla q > 0$? (and p9, l. 8.)

**Yes.**

p6, l. 2: "Between 4-8 km ... (Fig. 4g)." This is not clear from the figure.

**We have revised the manuscript, as that we observe dry conditions in the middle troposphere and relatively wetter conditions near the tropopause.**

p7, l. 2: "The ARM 95-GHz W-band ARM ..." one of the two ARMs seem redundant.

**Agree.**

p7, l. 32: "highlighting locations...ARSCL methods." It seems to me that the determination of cloud-top is improved by RWP, and cloud-base by MPL. Is this summary correct?

**Yes.**

p8, l. 16: " ... Table 2." Please see my first comment. It would be helpful to mention at this point that a variation of Table 2 is plotted in Figures 10 and 11.

**Ok.**

p8, l. 18: "Measurable precipitation (> 1 mm) ... light/trace precipitation (< 1 mm)." Is this accumulated precipitation over one day or one hour?

**This was in reference to the total daily precipitation. Fixed.**

p8, l. 24: "... below normal ..." A number representing the "normal" accumulated precipitation could be helpful.

**Fixed.**

p8, l. 27: "...convective and stratiform ..." Please see my third general comment.

**As above.**

p9, l. 30: "inspection of large-scale ... thermodynamics." Totally understandable statement, but isn't it true that the diurnal cycle of high-level clouds is tied to the diurnal cycle of the large-scale dynamics and thermodynamics through deep convection?

**Interesting comment. We have attempted to reword.**

p10, l. 13: "profile methods also distinguish columns with convective vertical air motions... as 'convective'." Please see my third general comment. And further details regarding the profiler methods could be informative.

**Added.**

p10, l. 24: "the difference in the mean rainfall rate are less pronounced, implying dry season convection as stronger (instantaneously), since the convective cell coverage is also reduced during the dry season ..." Recently Schiro 2017 (Thermodynamic Controls on Deep Convection in the Tropics: Observations and Applications to Modeling) examined animated radar data for the GoAmazon campaign period, and reached a similar conclusion.

**Thank you for the reference. Added. We assume you are referring to the dissertation,**

**Schiro, K. A.: Thermodynamic Controls on Deep Convection in the Tropics: Observations and Applications to Modeling, Ph.D. thesis, University of California, Los Angeles, Los Angeles, CA, 148 pp., 2017.**

p10, l. 28: "organized systems pass over T3 primarily in the morning hours during
the wet season..." Is this consistent with Figure 8(d)?

**Yes; The secondary peak in deep convective clouds between 23-03LT, is likely associated with propagating convection passing over T3 site.**

p10, l. 32: "...with the dry season having less organized cloud contributions..." Will Figure 9 be similar to Figure 8 if both figures are composed for only precipitating days?

**Unlikely. If we just consider the case of 'deep' cloud classifications, one might expect significant differences in the diurnal properties of those clouds simply owing to those from the wet season more likely having organized/MCS-type trailing stratiform components than the more-likely isolated convective dry season storms (having less stratiform components).**

p11, l. 3: "... stratiform..." Please see my third general comment.

**Ok.**

p11, l. 6: "... consistent with a response to increased surface heating and an increase in the surface latent heat flux ... " The increments are defined with respect to certain reference values, but it is not clear what these reference values are.

**This is unclear for the authors. Is the reviewer asking for reference values to the increased surface heating / surface latent heat flux in the morning hours as compared to other times of the day? Surface heat flux responds to solar forcing and has a similar diurnal cycle with downwards SW flux. In that sense, we expect both latent and sensible heat flux to increase from sunrise to local solar noon.**

p11, l. 13: "This pattern suggests ... under wet season conditions." If the shielding issue mentioned right after this statement is real, the pattern is also due likely to the suppressed contrast of cirrus cloud, isn't it?

**Our interpretation was that this implies that the contrasts we observe could be even more pronounced – that the wet season pattern could be missing some fraction of these cirrus clouds and the separation between the two seasons (if one was comparing longer term model regime statistic behaviors).**

p12, l. 22: "... sample sizes are potentially too small ..." Please see my first general comment. Including uncertainty/variability is a potential solution.

**As in response to those general comments.**

p12, l. 31: "...precipitating convective clouds..." Please see my third general comment. The definition of precipitating and non-precipitating clouds, i.e., the threshold, is not clear.

**As above.**

p16, l. 15: "...suggest clouds influenced by aerosol tend to have larger concentration of smaller droplets and fewer precipitation sized drops for clouds with similar LWC." With the evidence presented in the manuscript (no information about chemical composition, though seasonal wind direction might implicate), it is unclear that how the wet-vs-dry season contrast for larger-scale environment dynamic/thermodynamic conditions would affect the cloud-microphysical processes (e.g., collision-coalescence, precipitation scavenging, ... I suspect that this question can only be answered by later modeling studies), I would modify this sentence as something like "... is consistent with the hypothesis that clouds influenced by aerosol tend to have larger concentration of smaller droplets and fewer precipitation sized drops for clouds with similar LWC."

**Accepted.**

Figure 2(a), I suppose the hydrometeor frequency is defined with respect to a threshold. Assuming this is true, it would be nice to know the threshold.

**We use ARSCL cloud masking as part of this, which depends on the source of the dataset (e.g., WACR, MPL). For RWP, echoes with Z >= -10 dBZ are considered significant hydrometeor return. Both undergo filtering for significant echo based on other fields (for example, echo classification schemes to remove Bragg echo-dominated regions from the RWP)**

Figure 2(c), two additional horizontal lines could help the readers better capture the evolution of CAPE and CIN.

**We have attempted to modify several images as best as could satisfy multiple reviewer comments. This includes adding horizontal lines in Figure 2c for the corresponding mean value of CAPE and CIN to address the reviewer's suggestion.**

Figure 5(d), the rain rate is discretized with non-trivial units.

**Ok. Added to figure caption.**

Figure 7(b), it is not clear in the caption how a precipitating day is defined by the condition > 1 mm/hr. Does it mean the daily accumulated precipitation > 24 mm, or hourly accumulated precipitation > 1 mm for any hour of the day (also note p12, l. 28)?

**Correct; reference to daily precipitation, [mm], not an [mm/hr] threshold. Fixed.**

Figure 7(c), it is not clear that whether the fractions are calculated for all days or for only precipitating days.

**Relative to the total precipitation that was collected in those intervals; Thus, it would then only be applicable for precipitating days.**

Figure 10(a), the frequency of altostratus is 0 from 6 to 16 (assuming my reading of the color bar is accurate). Following the usual definition of conditional average, the frequency should be in the denominator. If this is how conditional SW CRE is defined, how could it be well defined for the same period when frequency is 0?

**The original figure had some errors in the white color. The altostratus frequency are quite low (< 1%) but not zero. The figure has been updated.**

Figure 12(f), The units of LWP are g/kg, which is different from the conventional definition and Figure 5(d). A typo maybe?

**Yes. Image has been corrected.**

**Response to Anonymous Referee #2**

*Overview*

*This manuscript presents a nice overview of unique datasets from the central Amazon, specifically focusing on vertical profile measurements of thermodynamic conditions and cloud boundaries with collocated surface radiative flux observations. It doesn't get into much detail on any one science question, but that is okay for an overview paper, especially since these datasets are not discussed in detail in Martin et al. (2016, 2017) overview papers. The results showing differences between wet and dry seasons are expected from previous studies, and the most interesting new results are the estimated cloud radiative effects of different cloud types as a function of season and diurnal cycle showing mean properties over a year that are similar to shortwave and longwave cloud radiative effects at a very rainy site in the Tropical Western Pacific. Another interesting finding is that cumulus clouds tend to be deeper with greater liquid water paths in the wet season than the dry season despite similar liquid water contents in the clouds and larger droplets in the wet season clouds. My primary concern is that the authors don't delve far enough into these interesting findings leaving the paper very descriptive with open questions. Being an overview paper of newly available datasets, I don't expect as much scientific investigation as in other types of papers, but I think that there are fairly simple and straightforward ways that the authors could attempt to explore some of the reasons for the aforementioned interesting results, as described further in major comments 5 and 6.*

**We thank this reviewer for their comments and suggestions, and we hope we have improved the revised manuscript in ways that respond to any concerns. As this is a combined response to several reviewers, we will attempt to respond directly to comments within this section, but may refer the reviewer to responses (and associated changes) to the other reviewers as well.**

*Major Comments*
*1. There are a lot of coauthors on this paper, and I can't help but wonder how each coauthor contributed to the paper. Does each coauthor meet ACP's guideline that "only persons who have significantly contributed to the research and paper preparation should be listed as authors"? What did each coauthor contribute?*

**We believe that the co-authors have all significantly contributed to the presented research and/or preparation of this manuscript. Overview efforts lend to acknowledging the many contributors required to collect, process, interpret the diverse insights covered by this manuscript.**

*2. The CAPE values seem extremely high. For example, they are far higher than those from Manaus in Machado et al. (2004) and in Williams et al. (2002) in Rondonia, Brazil (both papers that you cite). This could be a result of having mid-day soundings, but have you checked to make sure that these are correct? Was any comparison made with the 12 and 00Z Manaus operational soundings? And what is causing the CAPE to rise so dramatically to near 4000 most days? This is*

*an interesting feature. Is this confined to lifted parcels within a thin PBL layer, so that mixing erases much of it and prevents extreme convection from occurring?*

For the calculation of convective indices, we follow the methodology outlined in Jensen et al. [2015] and Moncrieff and Miller [1976]; CAPE is calculated as the vertical sum of the parcel buoyancy, calculated as the difference of the parcel virtual potential temperature and the environmental virtual potential temperature divided by the environmental virtual temperature (e.g., see eq. 1 in Jensen et al [2015]).

As the reviewer is likely aware, the calculation of CAPE (and CIN) is very sensitive to the choice of the originating parcel. For the purposes of this manuscript, we used two separate representations of the surface parcel characteristics. We calculate CAPE using the first sounding observations to represent the surface parcel (often referred to as the surface-based CAPE or SBCAPE from Bunkers et al. [2002]), and we also use the level of maximum virtual temperature within the lowest 1 km (this is similar to the "most-unstable" CAPE or MUCAPE from Bunkers et al. [2002]).

Bunkers, M., B. Klimowski, and J. W. Zeitler, 2002: The importance of parcel choice and the measure of vertical wind shear in evaluating the convective environment. Extended Abstracts, 21st Conf. on Severe Local Storms, San Antonio, TX, Amer. Meteor. Soc., P8.2. [Available online at http://ams.confex.com/ams/SLS_WAF_NWP/techprogram/paper_47319.htm.]

We note that these values often were the same; this approach tends to maximize the value of CAPE for each sounding. Without a quantitative way to determine the level from which a convective parcel originates (impacts of mixing), we choose a single origin for all soundings. Because of this sensitivity to the definition of the surface parcel, comparisons between CAPE values from different studies should be done carefully. We expect that much of this CAPE is never realized due to mixing.

*3. Increase the text size on Figure 3. It is currently unreadable. Also, the units in Figure 12a-b are off by 2 orders of magnitude.*

Agree. We revised Figure 3 to improve readability. Old Figure 12a-b concentrations were in logarithm scale, the figure has been revised in the revised manuscript.

*4. The hour to hour noisiness of rain rate in Figure 7 seems to indicate modest sample size, perhaps not surprising for a point location, even if observed for nearly 2 years. Does this impact the robustness of conclusions that require sub-setting of the data (e.g., diurnal cycle, seasonality)? The peak in rainfall at noon is also sooner than is typically observed in most land locations, and it doesn't seem to match up with the peak congestus and deep convection cloud fractions in Figures 6, 8, and 9. Is there a reason for this?*

As also described in our responses to Reviewer 1/3, we believe the results are consistent with studies that attempt to capture local T3 behaviors as compared to regional-domain behaviors (within a few hundred kilometer distance from the T3 site). For one example of how we have supported this further, this is demonstrated in ShCu comparisons against satellite observations, response to Reviewer 3.

For precipitation, we provide Reviewer 1 with the following image that demonstrates the standard deviation of instantaneous rainfall rates. We have also added reference in a new panel in Figure 7 to the total precipitation. There are fewer observations in the overnight hours, but also fewer instances of larger precipitation rates / convection. We agree with the reviewer if their comment is suggesting some variability is to be expected, simply based on having a single site for a two-year period.

[Figure]

As far as spatial representativeness, we do not agree that the peak we observe at T3 is uncharacteristic. Some confusion (as related to 'deep convective' cloud classifications) is that 'deep convection' cloud categories include the convective cells, convective lines, and the complement of MCS systems (i.e., recent MCS literature refers to as broad stratiform region, Houze et al. 2015). This includes what we consider as 'convective' precipitation, as well as stratiform precipitation components that trail those deeper convective cells.

Houze, R. A., K. L. Rasmussen, M. D. Zuluaga, and S. R. Brodzik, 2015: The variable nature of convection in the tropics and subtropics: A legacy of 16 years of the Tropical Rainfall Measuring Mission satellite. *Rev. Geophys.*, 53, 994-1021.

Our co-authors in Burleyson et al. 2016 (JAMC) have also shown that the frequency of deep convection around the T3 site begins to increase just before 1200 LT and has a broad peak across the afternoon during the rainy season (their Figs. 8, 10) which we still believe is consistent with our Fig. 8. The SIPAM S-band radar also shows measurable mean precipitation around the T3 site from 0900 – 1800 LT (their Fig. 11), which is consistent with our Fig. 7. Nevertheless, we do agree that some variability in Fig. 7 does arguably reflect some limitations with our sampling (esp. since overnight hours feature fewer instances of rain than afternoon hours, etc.) and some noisiness associated with rain gauge measurements;

*5. The similar CRE between Manaus and Manus is interesting, and perhaps expected, especially relative to Darwin and Nauru, which are not typical of many rainy, tropical places because of the dry air aloft that often impacts them. However, breaking down the overall SW and LW CRE at each site by frequency of cloud type and conditional CREs for each cloud type like in Table 2 would be much more interesting. Are some cloud types similar while others are different between sites? How do these similarities and differences impact the overall SW and LW CRE? Are there differences in the diurnal cycle of cloud type frequencies and CREs? Ideally, it would be extremely informative to investigate potential thermodynamic relationships with cloud frequency or CRE at the different sites or to pick a key cloud type like shallow cumulus and relate its CRE with its depth and LWP at each site. I'm not suggesting that I expect you to do all of this, but it seems that it would be fairly straightforward and not very time consuming to investigate a little more, placing this new dataset into better context with other well-observed sites.*

**The suggestions of this reviewer are interesting and valuable, and a detailed and extended analysis along these lines would be reasonable. However, the authors feel several items may represent a future step for GoAmazon datasets, insomuch that a level of care must be taken to perform the comparison rigorously so that one can understand any differences (rather than rushing this for a revision). As in other comment/responses, initially we found very low correlations between the cloud fractions and thermodynamical quantities of interest, in terms of direct or lag correlations, for various cloud types (e.g., perhaps even more time-consuming when the initial overview flow was not to drill down too heavily on a single cloud type, etc).**

**Further, digging into the comparisons between sites such as Manaus and Manus, there are interesting similarities and differences in SW CRE by cloud type (and we agree with modifying the manuscript to remind reviewers that the bulk similarities do not necessarily extend to individual cloud type behaviors). The authors note that results in Table 2 (individual cloud types values) can be compared with those from Burleyson et al. (2015) Table 4 (as provided below). For example, while the frequency of shallow clouds (mostly cumulus in these two regimes) is roughly similar, as is their average SW transmissivity, there are differences found in the mean SW and LW CRE for shallow clouds between the two sites.**

|  |  | Frequency (%) | Frequency as lowest cloud (%) | SW Trn | SW CRE (W m$^{-2}$) | LW CRE (W m$^{-2}$) |
|---|---|---|---|---|---|---|
| Low | Manus | 15.9 | 23.4 | 0.52 | −336.5 | 27.0 |
|  | Nauru | 12.2 | 20.9 | 0.58 | −291.4 | 26.2 |
|  | Darwin | 21.1 | 31.1 | 0.63 | −261.1 | 23.3 |
| Congestus | Manus | 5.8 | 8.5 | 0.36 | −432.7 | 30.8 |
|  | Nauru | 2.1 | 3.6 | 0.33 | −445.1 | 30.5 |
|  | Darwin | 3.7 | 5.4 | 0.33 | −451.9 | 34.3 |
| Deep convection | Manus | 6.7 | 9.9 | 0.19 | −539.5 | 33.1 |
|  | Nauru | 2.0 | 3.4 | 0.17 | −576.1 | 31.0 |
|  | Darwin | 5.7 | 8.4 | 0.16 | −560.2 | 34.9 |
| Altocumulus | Manus | 11.3 | 16.6 | 0.68 | −193.5 | 16.9 |
|  | Nauru | 5.0 | 8.6 | 0.70 | −178.3 | 14.6 |
|  | Darwin | 7.6 | 11.2 | 0.69 | −197.6 | 18.4 |
| Altostratus | Manus | 1.2 | 1.8 | 0.58 | −256.7 | 20.1 |
|  | Nauru | 0.4 | 0.7 | 0.55 | −292.3 | 19.3 |
|  | Darwin | 0.8 | 1.1 | 0.56 | −271.9 | 22.9 |
| Cirrostratus | Manus | 4.1 | 6.1 | 0.48 | −337.3 | 18.0 |
|  | Nauru | 1.4 | 2.3 | 0.50 | −329.8 | 14.8 |
|  | Darwin | 4.3 | 6.3 | 0.43 | −361.9 | 20.6 |
| Cirrus | Manus | 22.9 | 33.7 | 0.78 | −129.3 | 7.4 |
|  | Nauru | 35.1 | 60.4 | 0.83 | −100.1 | 5.3 |
|  | Darwin | 24.7 | 36.5 | 0.82 | −114.0 | 8.4 |

**Now, this could be due to a number of factors: variance in the clear sky downwelling SW or LW flux, the time of day when the shallow clouds are most prevalent, or the frequency of multi-layer clouds (something along the lines of thicker cirrus clouds occurring on top of the shallow clouds – more in Reviewer 3 response). Last to note, in our current study we define shallow cumulus clouds with cloud-top height < 3 km, whereas in Burleyson et al. (2015) the height threshold used was < 4 km. That implies the shallow clouds defined in our study are physically limited to being shallower than those in Burleyson's study (hence, optically thinner), potentially resulting in smaller SW CRE.**

**Identifying which of these factors are playing a role would help the community better understand the differences in cloud characteristics between the two sites; At present, we do think that this analysis is beyond the scope of what we intended for this overview paper. But, we do agree we should revise and mention that bulk agreement does not necessarily extend to all cloud types, and that these differences are an interesting topic for further study that should be highlighted in the revised manuscript. We have added some discussions of these issues in the revised manuscript.**

*6. Given the datasets available that are highlighted in this paper, it seems like it should be straightforward to investigate the cause of deeper cumulus clouds with greater LWP in the wet season as compared to the dry season, but with similar LWC. Is the cumulus cloud depth related to the CIN, relative humidity just above the boundary layer, both? When cumulus cloud depth is controlled for, do the dry season clouds have a stronger SW CRE caused by greater numbers of smaller droplets? How much of an increase in cloud depth is required to offset the cloud droplet SW CRE? These are questions that could probably be examined without too much effort that are important to understanding how all of these observations connect together through processes.*

**Again, a very interesting science question from the Reviewer, placing a different emphasis on another type of cloud regime within this diverse dataset. While the authors admit our response to this reviewer is possibly nonideal, we initially prioritized (in this set of responses) responding more to the full set of reviewer comments with respect to representativeness or spread of the observations, including additional comparisons against satellite observations. We believe those additions are more important (putting up a fence) as related to adding confidence in these datasets, so as to then be trustworthy as capable/appropriate for activities as outlined above (chasing the livestock).**

*7. ACP requires a "Data Availability" section on how all research data can be accessed, which goes before the "Acknowledgements" section. Please insert this section.*

**Added.**

*Minor Comments*

*1.On page 5, lines 5-7, I have trouble seeing higher CAPE and lower CIN in the transition periods in Figure 2. It looks fairly constant and the CIN appears lowest in the wet season and highest in the dry season, consistent with Figure 3, so can you clarify this sentence or better show it in the figure? It is then stated on page 11, lines 23-24 that wet season conditions favor weaker CAPE and dry season stronger CAPE, but the figure don't seem to show that. There is very little difference in Figure 3a, with enough spread that the differences between wet and dry season CAPE is likely not statistically significant.*

**As also to Reviewer 1, attempted to clarify. We were not clear about our usage of CIN (e.g., magnitudes). We agree that the differences (e.g., reports for slightly higher CAPE in the dry season, etc.) do not appear too pronounced (could be because of the use of radiosonde maximal daily values ); We have added horizontal lines (corresponding to mean values of CAPE can CIN) in Figure 2 to help show their variability with season. The behavior is arguably more pronounced for Fig. 3, but we have attempted to modify the text to a more conservative stance in line with the ARM observations (as compared to the expectations of previous efforts).**

*2.. On page 5, lines 16-17, it is stated that CIN decreasing during the day with CAPE peaking at mid-day is consistent with development of convection breaking the capping inversion and consuming CAPE, but I don't understand this argument. Isn't the layer of CIN (not necessarily an inversion, by the way) simply reduced through boundary layer mixing induced by daytime heating, while the same daytime heating warms the boundary layer and increases CAPE?*

**The argument would follow that the CAPE increases and the CIN decreases in the morning portion of the day as the surface is heated and the boundary layer is mixed. As convective parcels reach the level of free convection, CAPE is consumed such that later in the day the CAPE begins to decrease.**

*3. Is the moisture advection in Figure 4 the same as moisture convergence? How is it calculated? It is difficult to interpret 3-D advection. How much of the advection in Figure 4 is a result of the vertical component? Can it be stated if vertical mixing is the dominant component?*

**The 3-D moisture advection is the same as 3-D moisture convergence, since the difference between moisture advection** $-V_h \cdot \nabla_h q - \omega \dfrac{\partial q}{\partial p}$ **and moisture convergence** $-\nabla_h \cdot (qV_h) - \dfrac{\partial(\omega q)}{\partial p}$ **is** $-q\left(\nabla_h \cdot V_h + \dfrac{\partial \omega}{\partial p}\right)$ **which equals to 0.**

**We revised the text to add the calculation of total advection:**

*"Fig. 4 plots the diurnal cycle of the large-scale vertical motion (omega), total advection of moisture and relative humidity. The total advection of moisture is the sum of the horizontal and the vertical advections:*

$$q\_adv\_t = -V_h \cdot \nabla_h q - \omega \frac{\partial q}{\partial p}$$

*The total advection can be interpreted as 3-D moisture convergence. For the GoAmazon period, the vertical component dominate the total moisture advection (not shown). These large-scale fields were derived …"*

**The figures below show the horizontal (q-adv-h) and vertical (q-adv-v) moisture advections. The vertical component dominates the total moisture advection.**

[Figure]

*4. On page 7, it is stated that the RWP is used to reconstruct cloud boundaries up to 13 km, but really, isn't the RWP observing precipitation boundaries due to its sensitivity limitations? And because of these same limitations, doesn't it underestimate cloud top? These seems apparent in Figure 5, for example. If so, I recommend mentioning this limitation.*

**Agree. We have added some clarification to the text.**

*5. The CRE estimates depend on accurate estimates of the clear sky radiative fluxes. What causes the sharp decrease in clear sky longwave flux at 15Z in Figure 5f that appears coincident with the edge of the deep convective system? Is this accurate? If not, is there a bias in some situations that impacts longwave CRE?*

**The clear-sky LW flux estimates from the RadFlux product (Long and Turner 2008) was derived using actual detected clear-sky data when available, and uses surface measurements of air temperature and vapor pressure in the Brutsaert (1975) formula. As shown in Ohmura (2001) and later confirmed by McFarlane et al. (2013), the first 100 m and first 1 km above surface produce over 70% and over 90% of the clear-sky downwelling LW irradiance, respectively. Since this lowest portion of the atmosphere tends to be well mixed, one can use the surface 2m meteorological measurements to derive a relationship fairly well representative of the gaseous LW emission reaching the surface. Therefore, the drop in downwelling clear-sky LW is associated with the drop in 2m air temperature. But, the clear-sky downwelling LW would also decrease because of the large influence of the near-surface temperature and humidity on the clear-sky LW fluxes.**

**Brutsaert, W., (1975): On a Derivable Formula for Longwave Radiation from Clear Skies, Water Resour. Res., 11(3), 742– 744.**

**Long, C. N. and D. D. Turner (2008): A Method for Continuous Estimation of Clear-Sky Downwelling Longwave Radiative Flux Developed Using ARM Surface Measurements, J. Geophys. Res., 113, doi:10.1029/2008JD009936.**

**McFarlane, Sally A., Charles N. Long, Julia Flaherty, 2013: A Climatology of Surface Cloud Radiative Effects at the ARM Tropical Western Pacific Sites. J. Appl. Meteor. Climatol., 52, 996-1013. doi: http://dx.doi.org/10.1175/JAMC-D-12-0189.1 [pnnl-sa-89006]**

**Ohmura, A. (2001), Physical basis for the temperature-based melt-index method, J. Appl. Meteorol., 40(4), 753– 761.**

*6. It is mentioned that low level clouds impact the detection of upper level clouds by the micropulse lidar and ceilometer. Is this what caused the discontinuities in cirrus cloud fraction at 6 and 18Z in Figure 6a? If so, can this be stated? And does this mean that cirrus is significantly underestimated at the uppermost levels?*

**Underestimation of cirrus is a well-known issue. The drop in daytime cirrus detection by MPL is also likely affected by increase noise due to background solar flux. So, we suggest that it is a combination of low clouds and reduced SNR for the lidar when the sun is up. Below is the relevant text and citations from Burleyson et al. [2015] as related to the SNR issue:**

"One characteristic of the active remote sensors that needs to be considered is a reduced sensitivity to optically thin high-level clouds during periods of high solar elevation angles. As demonstrated in Comstock et al. (2002), there is an approximately 30% decrease in high cloud frequency detected by the MPL at Nauru during the 6 h period centered on solar noon when the sun is highest in the sky. This decrease is attributed to poor MPL signal-to-noise ratios when the solar background energy is large, increasing the noise. In addition, the ARM program physically covers the lidar telescope during times when the sun is nearly overhead to avoid damaging the receiver.

More recent comparisons between the MPL and Cloud-Aerosol Lidar and Infrared Pathfinder Satellite Observations (CALIPSO; Stephens et al. 2002) confirmed that the fraction of cirrus that is not detected by the MPL during daytime is ~25% for clouds above 10 km (Thorsen et al. 2013). Dupont et al. (2011) showed that almost 50% of situations over the ARM Southern Great Plains (SGP) site show a signal-to-noise ratio too low (smaller than 3) for the MPL to infer cloud properties higher than 7 km using the STRucture of ATmosphere (STRAT; Morille et al. 2007) methodology during summer daylight periods. These limitations suggest that any reduction in the frequency of occurrence of optically thin high-level clouds near solar noon may result from a sampling bias and thus must be interpreted carefully."

*7. On page 9, there are several places where the wording doesn't seem accurate:*
*a) On lines 5-6, it is stated that shallow clouds dominate the early morning hours, but to me, it looks like they continue well into the afternoon.*

**Adjusted.**

*b) On line 12, it is stated that congestus and deep convective clouds are prominent from mid-late afternoon, but it looks like they are prominent starting right at noon, and peak cloud fraction looks to be between 12 and 1 PM for congestus and 12 and 2 PM for deep convection rather than at noon, as is stated in the paper.*

**Adjusted.**

*c) On line 21, it is stated that the convection aligns with the mid-upper level vertical motion, but there is a secondary maximum in deep convection overnight when there is descending mid-upper level motion, so these don't seem perfectly aligned to me.*

**Adjusted.**

*d) On line 24, the pre-dawn peak in altocumulus is its primary peak, not its secondary peak.*

**Adjusted.**

*e) Lines 25-27: The wording here is confusing. What does precipitation have to do with congestus cloud fraction peaks? Please clarify.*

**By our definitions, the congestus clouds are associated with a nontrivial fraction of the precipitation.**

*8. Stating that 103 wet season days produce 1600 mm of rainfall and 52 dry season days produce 600 mm of rainfall doesn't necessarily lead to a factor of 2 difference in average rain rate. I can see that differences in Figure 7, but the argument from the number of days and precipitation perspective requires knowledge of how many total wet season and dry season days were sampled.*

**Agree.**

*9. The statements on page 10, lines 5-7 and lines 23-25 seem contradictory with one saying that relative convective intensity is not much different between wet and dry seasons, while the other says that dry season convection is stronger. I personally don't see evidence in Figure 7 that dry season rain rates are more intense with conditional rain rates in both seasons that are similar.*

**The reviewer is correct that nothing presented in Figure 7 (original, or revised) would immediately demonstrate this concept on its own. The key statement in the manuscript was that "...the overall convective cell coverage is also reduced during the dry season (e.g., Giangrande et al. [2016])." This was based on area/temporal convective coverage from that RWP (e.g., same dataset, location, etc.).**

*10. On page 10, line 28, how are you defining "organized systems"? And similarly, on page 11, line 1, how is "organized cloud" defined?*

**Changed references to mark these as mesoscale convective systems.**

*11. It's not surprising that clouds and thermodynamics are not strongly correlated when viewed from a stationary vertically pointing perspective, but this shouldn't be confused with them not being correlated over a larger scale from a Langrangian perspective. When you state that weak correlations are found between cloud behaviors (isn't cloud state a better term here than behavior?) and thermodynamic parameters, what time scale is this correlation being computed on and is it a time lag correlation?*

**We agree with cloud 'state' and have changed the wording. The initial reference, Collow et al. [2015] and its reference to Kollias et al. [2009] (and associated references within that) as best as we can determine did not factor in the lag correlations within a given day. When we performed our checks on these ideas starting from Collow's reference, we did adjust our calculations to determine the impact for considering different sounding launches (relative to these diurnal variability), but not necessarily ensuring these were always 3h before the time of cloud observations/initiations, 6h before cloud initial observations, etc. Nevertheless, those forms of lag correlations also suggested that nothing generally 'jumped out' as far as significant**

improvements to those correlations between quantities of interest and sounding parameters. This also relates back to a previous Reviewer 2 general comment, e.g., it is questionable that there are necessarily easy/quick relationships between the thermodynamics and the cloud fractions.

*12. On page 12, line 10, I believe you can delete "Compared to SW CRE" to make the sentence read more clearly. This sentence mentions SW CRE being much larger than LW CRE, but SW CRE is 0 at night and the surface energy balance is much different at night than during the day, so a given LW CRE at night, even if small compared to the larger daytime SW CRE, could have just as significant of impacts on variables such as surface temperature, couldn't it?*

**Yes. We have deleted those words. LW CRE at night has a much larger effect on surface energy budget compared to daytime.**

*13. On page 13, line 2, insert "and" between "(SW)" and "longwave" with a comma after "fluxes". And on line 29, add an "s" onto "peak".*

**Ok.**

*14. On page 15, line 15, it is stated that sharper CAPE and CIN contrasts during the wet season and transitional periods enhance the likelihood for deep convection to have organized components. First, how are organized components defined? Second, how is this known? I didn't see any evidence presented that shows this and important components of mesoscale convective system growth such as vertical wind shear were not discussed. Mid-upper level humidity and vertical motion are also potentially important factors aside from CAPE or CIN.*

**The reviewer is correct. We did not present any evidence in this particular paper that MCS systems are more likely to develop in these transitional regimes. We have modified this line.**

*15. Consider rewording "supports local congestus to deeper cloud triggering" on page 15, line 19.*

**Ok.**

*16. On page 16, line 1, consider rewording to "between maritime-like 'active monsoon' conditions with widespread clouds and precipitation and continental-like 'break monsoon' conditions with less clouds and precipitation, but more intense deep convection".*

**Ok.**

*17. On page 16, line 10, insert "cumulus" after "thicker".*

**Ok.**

*18. What is meant by "natural cloud laboratory" on page 16, line 14?*

**Removed.**

*19. The last paragraph on page 16 has some confusing wording in spots. For example, the third sentence seems out of place in the paper since the Manaus plume was not discussed at all in this study. Additionally, the last two sentences don't seem worded well. Can you attempt to clarify what is meant in these sentences?*

**Adjusted slightly in response also to Reviewer 1.**

**Response to Anonymous Referee #3**

*Overview*
*This manuscript describe a unique long term set of observations, as well as aircraft IOPs, in the Amazon. These measurements are necessary to help better constrain global climate models and parameterizations of clouds and precipitation for regions, like the Amazon, that have been challenging to simulate. Due to the long term nature of the observations they are able to look at cloud and precipitation over the diurnal and seasonal time scales. Their observations of clouds, are limited from these long term sites due to the nature of their 1D observations. From an aircraft based background or a satellite perspective, these estimates of cloud fraction aren't ideal. The works isn't groundbreaking, but it is a good overview paper of the observations from the project that aren't addressed in the other GoAmazon papers.*

**We thank the reviewer for their comments and perspective. We hope our revisions/additions help clarify several concerns.**

*Main Comments*
*1. How often are multi-layer clouds observed? The authors discuss the fact that if they removed the multi-level cases they wouldn't have enough data for analysis. This is a concern when thinking about sorting data by convective and stratiform. Are these categories meaningful if it's only based on the lower most data? What if there are both convective (low level) clouds and higher level sratiform clouds that are obscured? Does this have an impact on the rain rate? Are the rain rates difference from convective clouds only vs. multi-level louds?*

**We note that for precipitating / deep clouds, the presence of multi-layer clouds are not a major concern for our definitions [since deep clouds already occupied the column between low (<3km) and upper (> 8km) atmosphere]. Now, we have examined the frequency of single- and multi-layer clouds (regardless of their cloud types) for the entire GoAmazon dataset, and breakdown this by wet and dry seasons (see the figure below). Over the entire field campaign period, single layer clouds occur ~48% of the time, whereas multi-layer clouds occur ~20% of the time. During the wet seasons, multi-layer clouds occur twice more frequently than during the dry seasons (~28% vs. ~11%).**

**Now, we suppose the Reviewer's initial comment was in response to our sentence (near the end of the second paragraph in section 4):**

**"*However, we also note that it is not possible to separate the radiative impact of multi-layer clouds and sample sizes are potentially too small to only consider single-layer cloud periods.*"**

**What was meant was that it was difficult (sample size) to consider times when we separate by cloud types and then further by hour of the day to estimate their conditional CRE. As seen in the sample size figure in our response to your Comment 2, this includes both single- and multi-layer clouds. Some cloud types (e.g., altostratus, cirrostratus) have rather low sample sizes**

during certain hours of the day. Further separating single- vs. multi-layer may result in limited samples, not representative of the radiative fluxes associated with those cloud types. We did attempt to examine the potential impact in estimating conditional SW CRE when using single- vs. multi-layer clouds.  In section 4.1 second paragraph, we wrote:

"*Examination of the averaged conditional SW CRE calculated using only single-layer clouds reveal a relative reduction of ~26% for altocumulus and ~20% for shallow cumulus clouds, and negligible difference in other cloud types.*"

While we could choose to use only single-layer clouds, we explained the reason for including multi-layer cloud periods in estimating CRE in the same paragraph:

"*Note, the conditional CRE presented in Table 2 includes both single-layer clouds, as well as when additional cloud layers are above the lowest detected cloud layer. This is done deliberately to be consistent with the method used by Burleyson et al. [2015] such that the GoAmazon2014/5 results can be directly compared with their long-term results from the ARM TWP sites.*"

[Figure]

*2. Estimates of uncertainty are missing and should be addressed. Also, on several figures an idea of the sample size would help put the data into context. It is difficult to evaluate the data when it is unclear how much data is actually included in the figures.*

**As also in response to Reviewer 1 and 2, we have added references to the degree of uncertainty / spread in some images, tables and the text. Please refer to our detail response to Reviewer #1's first question, which is similar to the uncertainty/variability question this reviewer has asked. In terms of the sample size, we produced a figure below that shows the total number of samples going into calculating the averaged frequency of occurrence of the lowest cloud type. As it can be seen from the figure, all clouds are fairly well sampled (if both single- and multi-layer clouds are considered). In the dry season, there are a few hour bins for deep convection, altostratus and cirrostratus that have insufficient data sample (< 5 days with those clouds appearing as lowest cloud type at those hours), shown in white colors.**

**In some cases, the authors believed the number of events/data were self-explaining, in that it was limited to the same number of days as those in the campaign, etc., with the overwhelming majority of those days featuring clouds and/or precipitation.**

[Figure]

**Figure: Number of samples used in calculating average frequency of occurrence of the lowest cloud type for each hour. White bins are hours with less than 5 data samples.**

*3.I'm confused why they bring in the other locations (Darwin, Nauru and Manus)? These data are not discussed anywhere else in the paper except in section 4 and to add a paragraph in section 5. It seems out of place, it could be part of another paper.*

**Simply, one motivation was that this dataset that allows new breakdowns of cloud types and contributions to surface radiation budget in the tropics; This is still a very rare observation; Our study provides an opportunity to contrast these results to others at other tropical locations. Or, one of the overarching goals of GoAmazon was to place an additional marker in the tropics for climate studies, particularly for an undersampled region. By comparing/contrasting the CRE effects in the Amazon with those in the Maritime Continent, we may quantify, for example, to what degree a cumulus cloud in one location is similar to a cumulus in another.**

*4. Have the authors considered using MODIS CF to get an idea how well their CF estimates match satellite observations? They can also use CALIOP to see how their estimates of multi-level cloud classifications compare to the ground measurements. Finally, they can use TRMM or GPCP data to get regional estimates of precipitation. These would put their work into a larger scale context.*

**As one of the larger additions/responses to this manuscript review, we have performed a comparison between GOES/VISST, MODIS for shallow cumulus (see two figures below). One of these images, and a new discussion, have been added to the revised manuscript as a new sub-section 3.3.**

**GOES VISST/(now SatCORPS) and MODIS datasets were available from Feb-Dec 2014. The GOES product has a 4 km resolution, 30 min update, and the MODIS product has a 1 km resolution and two overpasses. ShCu definition is set as a pixel-level cloud-top height < 3 km (fractions calculated at various spatial coverage domain).  Our results show that GOES, MODIS underestimate ShCu compared to T3 (second set of images below), with a larger bias in the wet season than in the dry season. We believe the main reason for this discrepancy is likely the presence of multi-layer clouds obscuring the detection of ShCu from passive satellite sensors (related to a Reviewer comment as above).  Spatial mapping of ShCu CF (upper image) suggests that T3 is representative of large area in the central Amazon basin.**

**GOES and MODIS:**

[Figure]

**Key points:**

- Wet Season: Feb 1 – May 1, Dry Season: July 1 – Oct 1
- GOES estimate mean ShCu fractions similar regardless of area.
- MODIS 1 km data provides slightly higher ShCu CF estimates.
- Values in the legend are averaged cloud fraction during daytime (07-17 LT)

**GOES, MODIS, w/ T3 overlaid (New Figure 9):**

[Figure]

**Key Points:**

- GOES underestimate ShCu fraction compared to T3.
- The peak timing of GOES ShCu is roughly comparable to T3.
- MODIS 1 km data provides slightly higher ShCu CF estimates.

*Minor Comments*
*There are often generalizations and wording issues that make their points less clear. These issues can be fixed by modifying their text. See specific comments below:*

*Page 2 Line 30 – "cloud study complement to the ... " → "cloud study to complement the..."*

**Ok.**

*Page 3 Line 3 – "cloud types and contrasts" what do you mean by "contrasts", do you mean differences in atmospheric conditions and thermodynamics, seasonal or diurnal changes. This is vague.*

**Fixed.**

*Page 3 Line 4-5 – Wording: "This analysis includes additional relationships to campaign aircraft in-cloud observations when available." → "This analysis includes additional relationships between in-cloud aircraft campaign observations when available."*

**Ok.**

*Page 3 Line 10-11 – What does "possibly maritime-like atmospheric conditions" mean, this is a vague comment and needs clarification.*

**Dropped.**

*Page 3 Line 12 – Clarify what you mean by a region of "underlying moisture." How would you define this, humidity, PW?*

**(humidity)**

*Page 3 Line 16 – "work has found a robust relation … " → "work has found a robust relationship …"*

**Ok.**

*Page 3 Line 19 – Clarify "cloud lifecycle complexity" what complexities are you suggesting are additional versus not-additional, the wording here is unclear.*

**Fixed.**

*Page 3 Line 34 – What are the "environmental forcing data sets?"*

**Modified as also in response to Reviewer 1.**

*Page 4 Line 8 – The pencil-beam/soda-straw description is not necessary.*

**Dropped.**

*Page 4 Line 9-10 – Please elaborate on how CF is described, so 50% cloud cover is recorded when there are cloud present for 30 minutes out of an hour? This seems strange compared to thinking of CF as a fraction of an area if I am interpreting the description correctly.*

**Yes; For ARM vertically pointing measurements, this is typically done to compare with satellite area cloud fraction estimates (e.g., Xi et al. 2010). It is fairly standard practice within the ASR/ARM DOE and associated climate modeling community.**

**Xi, B., X. Dong, P. Minnis, and M. M. Khaiyer, 2010: A 10 year climatology of cloud fraction and vertical distribution derived from both surface and GOES observations over the DOE ARM SPG site. *J. Geophys. Res.*, 115, D12124.**

*Page 4 Lines 17-18 – Does this choice, of using the maximum virtual potential temperature, result in a bias towards higher CAPE? Did you try a max, min and mean virtual potential temperatures to see how this changed the results?*

**As also in response to other Reviewers, we attempted two separate commonly used definitions of the surface parcel. We use the max virtual temperature in the lowest kilometer, which is similar to the "most unstable" CAPE (MUCAPE; Bunkers et al. 2002) and we used the lowest sounding observation, or "surface-based" CAPE (SBCAPE; Bunkers et al. 2002). We find that often the value of MUCAPE and SBCAPE are the same (i.e., the max Tv is at the surface). For the soundings analyzed in this study: SBCAPE and MUCAPE are equal for 56% of the soundings, SBCAPE > MUCAPE for 24% and MUCAPE is larger for 17%. However, we agree we should specify this as 'MUCAPE' in the revised manuscript.**

*Page 4 Lines 32 – wording is strange for "on their suitability" I suggest rewording this sentence.*

**Ok. Fixed.**

*Page 4 Line 32 – Page 5 Line 1 – What is shown in panel a) of Figure 2. The text here suggests cloud fraction, but the label says "hydrometeor frequency." Clarify.*

**All "cloud fractions" in the paper can also be considered "cloud frequency", e.g., it is simply a count of cloud occurrences divided by the total number of valid observations within an hour. We have clarified this in the manuscript. The label in Figure 2a has also been changed to "Cloud Frequency" to be consistent with other manuscript figures.**

*Page 5 Line 7-8 – Referring to Figure 3, it is not obvious in the CAPE panel that there is a difference between the wet and dry periods. They look the same based on Figure 3a.*

**A few reviewers have mentioned this and we have scaled back our language with respect to these differences and their significance.**

*Page 6 Lines 27-28 – How are the SCMWF analysis outputs constrained using the surface rainfall? The constraints are not identified in the text. Is it precip or no-precip? A certain amount of precip with a specific threshold? Clarify.*

***Text has been modified to state:***
**"These large-scale fields were derived from the ECMWF analysis outputs over the entire field campaign using a constrained variational analysis method of Zhang and Lin [1997]. The upper-level state variables (wind, temperature, moisture) from ECMWF are adjusted to conserve column-integrated mass, moisture and energy. Surface rainfall rate from the SIPAM radar is used as a major constraint. Details of these large-scale fields for the GoAmazon2014/5 deployment can be found in Tang et al. [2016]. This variational analysis was performed…"**

*Page 7 Lines 31-32 – (also noted in the figure comments) – where is the red bars located? Are they the thin lines above panel c? If so, a new way to note this should be found, it is not clear or easy to use these bars for the purposes described in the text.*

**We increased the width of the color bars above panel (c), and replaced red color (RWP period) with magenta color to enhance readability. Additional figure caption was added to describe these details.**

*Page 8 Lines 5-6 – Why are there no stratus or stratocumulus categories? The only stratiform clouds are altocumulus and cirrocumulus? Are there just so few of these cloud categories that you are leaving them out?*

**We are assuming the reviewer is asking about "low-level stratocumulus/stratus clouds". Our cloud type classification algorithm only uses cloud base/top boundary heights to differentiate cloud types. If there are stratus/stratocumulus clouds, they will all be classified as shallow clouds. In reality, stratus/stratocumulus clouds rarely exist at T3 during GoAmazon, because persistent low clouds (i.e., no break in time for many hours) were simply not observed in our dataset. The frequent gaps in time for low clouds are consistent with the nature of shallow cumulus clouds. Therefore, we denote low clouds as primarily shallow cumulus in this study. For trailing stratiform rain (e.g., behind a convective line in a MCS type system), these regions are included in 'deep convective' cloud.**

*Page 8 Lines 24-25 – You bring up that the 2014-2015 rainy season maybe be different than climatology. Perhaps it would be beneficial to show a monthly climatology for a long period of time to which you can compare the 2014-2015 rainy season? This way the readers can know how different this particular year is from the climatological average. After reading this comment I was left wondering if these results are just a special case or if they are in fact representative of this region in a general sense.*

As also in response to other reviewers, this statement was partially influenced by another published GoAmazon2014/5 study that made this particular claim (we acknowledge their efforts as a reference of work ongoing from GoAmazon). We have included the value for a 'normal' year. We also highlight Burleyson et al. [2016] who investigated this (and the representativeness of T3 spatially), as also in our previous response to Reviewer 1, as in that response:

"The topic was a focus of a related GoAmazon2014/5 effort by some of our coauthors – Burleyson et al. 2016 (JAMC). Their primary finding was that the diurnal cycle of deep convection around the GoAmazon sites is a superposition of locally forced convection and the inland propagation of the previous day's sea-breeze front. Their climatology showed that deep convection begins around noon around the sites, and has a broad peak in the mid-afternoon over the T3 site. There is some variability among the various GoAmazon sites that is related to localized circulations generated by the river breezes.

More to the Reviewer's original comment, the diurnal cycle at the T3 site is generally well correlated with the region to within a few degrees of the site, but the correlation drops off sharply at distances more than a few hundred kilometers. This is particularly important during the wet season. We would encourage the Reviewer (or those reading this response) to download that paper for more information."

*Page 8 Lines 25-27 – This would be a good place to include a more detailed description of how you calculate the precipitation as a function of convective and stratiform clouds. This is difficult and the method isn't clearly stated.*

We have attempted to improve some of these descriptions, also in response to Reviewer 1.

*Page 9 Line 19 – What do you mean by "rare?" How often do these sea breeze intrusions occur? How many times did this happen over the study period? What impact do these intrusions have on the results?*

We removed this term and description (was a poor interpretation by the lead author from a few co-author suggestions). A few of our co-authors (Machado, Burleyson, Schumacher) are currently working on efforts to better communicate these squall line statistics (satellite or radar-based). In some cases, these (and other) authors have already shown that the number of squall lines per year may certainly vary as function of the distance from the coast. Specific to T3, we would not think of it as commonplace for squall lines to reach Manaus (e.g., sampling the same squall line). However, complicating this is that these squall lines, although they may dissipate before reaching Manaus, might argue that some instability or humidity continues to propagate over the site. This added complexity is certainly suggested in looking at the Burleyson et al. [2016] composites.

*Page 10 Line 8 – Please describe how you separate the rainfall rates in to convective and stratiform types. Is this based solely on the cloud classification (as shown in Figure 5 c)?*

**We have modified text in these sections. These breakdowns are separate from the cloud classifications, in that 'deep' convective clouds will potentially contain 'convective' and 'stratiform' rainfall contributions. For example, the figure below (Fig. 3 in Giangrande et al. 2016) shows an MCS event from the RWP dataset. "Stratiform precipitation" is defined between ~13:20 UTC to ~17:10 UTC, where a clear bright-band signature is observed; "convective precipitation" is defined between ~11:20 UTC to ~13:20 UTC. But in the cloud type classification, both convective and stratiform precipitation are categorized as "deep convective clouds" as their echo-top heights both exceed 8 km and thickness > 5 km.**

[Figure]

**Figure 3.** Example (a) reflectivity factor *Z* measurements and (b) vertical velocity retrievals from the 20 March 2014 event during GoAmazon2014/5.

*Page 11 Line 3 – "stratiform precipitation (identified as "Deep Convection"..." this is confusing. Do you mean stratiform as in cirrostratus? Usually when I see the term stratiform I think of stratus or stratocumulus. Clarify this section.*

**We have modified some text, but in this context, it would refer most commonly to the trailing precipitation shield behind organized/MCS convective systems – e.g., those regions having pronounced bright band / radar signatures from aggregation/melting.**

*Page 12 Line 14 – The "green ocean" comment again, it's not really fully discussed in the beginning section (Page 3 Line 10) where it is first mentioned so it's strange to mention it again.*

**We have better attempted to reference to the origins of the term in the upper parts of the manuscript. The concept is one of a so-called hybrid environment for clouds that promotes both continental and tropical type characteristics.**

*Page 12 Line 28 – "better" this is not a descriptive term. What makes it better? Be more specific.*

**Ok.**

*Table Comments*

*Table 1-3 – The format of these tables is difficult to read in the current format. I'm assuming that they will be formatted differently when published.*

**We modified the table formats to make them fit in a single page for easier reading for the reviewers. The final format of the table is up to the journal.**

*Table 3 – Why are the authors including these other locations that are very far from the Amazon. When reading the text, there doesn't seem to be a solid justification for this other than to make a quick comparison.*

**Observations of this sort are rare. There are few references for tropical behaviors. This also ties back into previous 'green ocean' claims by other authors.**

*Figure Comments*

*Figure 2 – Continue the IOP dashed lines all the way to the top of the figure through panels a, b, and c. Perhaps make solid lines to section off the wet and dry periods as well. This would be helpful for knowing where the cut offs are. I drew them on myself to make it easier for me to see when they started and ended. What is "hydrometeor frequency" shown in panel a). In the test it suggests CF (Page 4-5 Lines 32 -33 "The more pronounced shifts during the wet season include increased CF in the mid0to- upper troposphere (between 3-10 lm, Fig 2a). So is it CF or something else that is being shown as frequency. Please clarify.*

**CF and cloud frequency are interchangeable for this study and this has now been noted in the revised manuscript. We extended the IOP dash lines across all panels as the reviewer suggested, along with some additional modification to the figure suggested by other reviewers to improve readability.**

*Figure 3 – Axes labels are small and hard to read. What is going on with the 19 am data for CIN. Why are they essentially a point?*

**For the 14 UTC sounding time the vast majority of soundings have a CIN value of zero (38 of 56 for the wet season and 34 of 39 in the dry season). When CIN is not equal to zero, the values are generally very small. Figure has also been modified for text sizing.**

*Figure 4 – Hard to read the panel labels for (g), (h), and (i). Perhaps move them outside and next to the left side of the figures for all the panels.*

**Figure has been modified.**

*Figure 5 – It is difficult to read the label or panel c), it is obscured by the cirrus clouds. The green and red line above panel c), is that the red bars referred to on Page 7 Lines 31-32. If so, these are near impossible to see clearly.*

**We added a white background to the label in panel (c). We also increased the width of the color bars above panel (c), and replaced red color (RWP period) with magenta color to enhance readability. Additional figure caption was added to describe these details.**

*Figure 6 – The white contours (starting at 10%) are difficult to see, perhaps increase the line thickness.*

**We increased the thickness of the white contour lines for Figure 6 and new Figure 8 (combining previous Figure 8, 9 - as suggested by the reviewer).**

*Figure 7 – For a) How many days are included (are there equal number of days in all the time bins?) b) same as a), how much data is included? c) Define the fractional accumulation more clearly (convective/stratiform). Does the number of samples change for each time bin? Are some time bins 100 samples (80 wet 20 dry) while others are 10 samples (8 wet 2 dry). It would be nice to know how much data is going into theses curves.*

**We have modified this plot to include a total accumulation plot. When attempting to respond to this, we believed this to be more useful than a plot showing # of instantaneous rainfall rate samples (e.g., counts of nonzero rainfall rates, etc.).**

*Figure 8 & Figure 9 – Perhaps these two can be merged so you can easily compare the difference between the wet and dry seasons. A difference panel, or just showing the difference between wet and dry would be a good way to show were the differences are most pronounced. As with Figure 6, the white contours (starting at 10%) are difficult to see, perhaps increase the line thickness.*

We thank the reviewer for the good suggestion. Figures 8 and 9 were combined into a single new Figure 8. We kept the order of the cloud types the same with Figure 6, but put wet and dry seasons next to each other to assist in comparison.